 **eLIFE**

# Super Spy variants implicate flexibility in chaperone action

**Shu Quan[1], Lili Wang[1], Evgeniy V Petrotchenko[2], Karl AT Makepeace[2], Scott Horowitz[1], Jianyi Yang[3], Yang Zhang[3], Christoph H Borchers[2], James CA Bardwell[1,4]\***

[1]Department of Molecular, Cellular, and Developmental Biology, Howard Hughes Medical Institute, University of Michigan, Ann Arbor, United States; [2]Department of Biochemistry and Microbiology, Genome British Columbia Proteomics Centre, University of Victoria, Victoria, Canada; [3]Department of Computational Medicine and Bioinformatics, University of Michigan, Ann Arbor, United States; [4]Department of Biological Chemistry, University of Michigan, Ann Arbor, United States

**Abstract** Experimental study of the role of disorder in protein function is challenging. It has been proposed that proteins utilize disordered regions in the adaptive recognition of their various binding partners. However apart from a few exceptions, defining the importance of disorder in promiscuous binding interactions has proven to be difficult. In this paper, we have utilized a genetic selection that links protein stability to antibiotic resistance to isolate variants of the newly discovered chaperone Spy that show an up to 7 fold improved chaperone activity against a variety of substrates. These "Super Spy" variants show tighter binding to client proteins and are generally more unstable than is wild type Spy and show increases in apparent flexibility. We establish a good relationship between the degree of their instability and the improvement they show in their chaperone activity. Our results provide evidence for the importance of disorder and flexibility in chaperone function.

**\*For correspondence:**
jbardwell@umich.edu

**Competing interests:** The authors declare that no competing interests exist.

## Introduction

Despite years of intense effort, the precise mechanism by which chaperones interact with proteins to enhance their folding is not entirely clear. We reasoned that we might gain insight into this long-standing problem by isolating and characterizing chaperone variants that exhibit improved chaperone activity. A genetic selection that we had developed previously gave us a unique opportunity to pursue these aims. This selection uses a folding biosensor to directly link protein stability to antibiotic resistance. The biosensor consists of an unstable protein inserted into β-lactamase, a selectable marker that encodes penicillin resistance (*Foit et al., 2009*). Stabilization of the unstable protein results in higher levels of antibiotic resistance. We showed that the stabilization could be due to mutations within the unstable protein itself (*Foit et al., 2009*), addition of chemical chaperones to the growth media (*Hailu et al., 2013*), or host variants that stabilize the unstable protein (*Quan et al., 2011*).

We isolated host variants that greatly stabilize poorly folded variants of immunity protein 7 (Im7), increasing their steady-state concentrations in the cell. We found that this stabilization occurs through the induction of a previously uncharacterized chaperone called Spy (*Quan et al., 2011*). We obtained evidence that Spy acts in an ATP-independent manner to help protect bacterial cells from a number of conditions that lead to widespread protein denaturation and aggregation, such as treatment with tannin, ethanol, or butanol (*Quan et al., 2011*). The crystal structure of Spy shows that it forms an unusual cradle shaped dimer (*Figure 1*; *Quan et al., 2011*; *Kwon et al., 2010*). When we attached environmentally sensitive probes to various sites in Spy, including the concave and convex surfaces, nearly all showed substantial changes in fluorescence upon interaction with the client protein casein.

**eLife digest** Proteins are made from long chains of smaller molecules, called amino acids, that twist and fold into complex three-dimensional shapes. Folding into the correct shape is crucial for a protein to function properly because many proteins work by binding to certain other proteins or molecules, like a key fitting into a lock. Additional proteins called chaperones often help with this folding process, and it has been proposed that chaperones must be particularly flexible in order to cope with the changes in the shape of the different proteins being folded. However, studying this hypothesis directly has proven to be difficult.

Now, Quan et al. have tackled this challenge by using a bacterial assay—that they had developed previously—and which links the correct folding of a test protein to cell survival and growth in the presence of an antibiotic. This approach was formerly used to identify a new chaperone called Spy, and Quan et al. have now used it to find variants of this protein that perform as even better chaperones. This assay identified several variants of Spy that could stabilise an unstable test protein even more effectively than the wild-type Spy can. All of these variants were also better than the wild-type Spy at stabilising two other unfolded proteins—and so were dubbed 'super Spy' proteins.

The mutations in the super Spy variants altered a region on the surface of Spy, which additional experiments revealed was likely to be involved in binding to the partner proteins. Furthermore, prior to binding to these partner proteins, the super Spy variants appear more flexible than the wild-type Spy protein. Quan et al. suggest that this increase in flexibility allows the super Spy variants to bind more tightly to a range of substrates, thus optimising their chaperone function.

These results suggest that client binding may occur over large regions of Spy, that Spy might undergo significant conformational changes upon client binding, or a combination of both (*Quan et al., 2011*).

We decided to investigate the mechanism by which Spy interacts with proteins and learn more about Spy's properties as a chaperone. We used a genetic selection similar to that used to discover Spy in an attempt to further enhance Spy's chaperone properties. We have now isolated Spy variants with improved ability to stabilize a poorly folded client protein (Im7 L53A I54A) in vivo. These variants also showed improved ability to prevent the aggregation of client proteins in vitro. Many of these Spy variants contain residue substitutions that act to expand a hydrophobic region present on the protein's concave surface. Crosslinking and hydrogen-deuterium exchange measurements suggest that this hydrophobic region is involved in client protein interaction. Our optimized Spy variants bind the client protein Im7 more tightly than wild type Spy does but are generally less stable suggesting that flexibility is important in the function of Spy as a chaperone.

## Results

### Identification of Spy variants with improved ability to stabilize a poorly folded client protein

We expressed a protein stability biosensor in an *Escherichia coli* strain that co-expresses the gene for the chaperone Spy under the IPTG inducible Trc promoter. The stability biosensor consists of a tripartite fusion that contains the unstable protein Im7 L53A I54A inserted into β-lactamase under the constitutive β-lactamase promoter (*Foit et al., 2009*). This partially unfolded variant of Im7 was chosen because Spy overproduction is known to stabilize it in vivo (*Quan et al., 2011*). Increasing the expression level of the chaperone Spy by increasing IPTG concentrations results in improved penicillin resistance encoded by the β-lactamase-Im7 L53A I54A biosensor (*Figure 1—figure supplement 1A,B*, focus on wild-type [WT] traces [black lines]). We reasoned that if mutations in Spy increase its specific activity as a chaperone, they should also be capable of enhancing the stability of the biosensor and thereby also enhance antibiotic resistance.

Our ability to link protein folding to antibiotic resistance gives us a unique opportunity to select for activity-enhancing mutations in a chaperone. Analysis of the reasons behind the improved chaperone ability of activity enhancing mutants of Spy should inform us about Spy's catalytic mechanism and perhaps also tell us what makes for a good chaperone. We reasoned that activity-enhancing mutations would be more informative in general than those that decreased function, in part because there are a wider variety of uninteresting reasons that mutations can disrupt function such as those causing chain

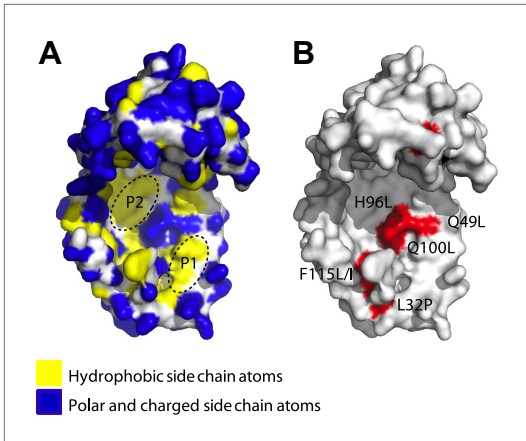

**Figure 1**. Surface presentations of the crystal structure of Spy (PDB ID: 3O39). The majority of activity-enhancing mutations localize to areas adjacent to hydrophobic patches. (**A**) Surface properties of Spy. Backbone atoms are shown in white, hydrophobic side chain atoms in yellow, and polar and charged side chain atoms in blue. Black dashed lines circle the two predominant hydrophobic patches P1 and P2. (**B**) Sites accommodating beneficial mutations. Side chain atoms of the residues identified as mutations in the genetic selection are shown in red. Q25 is in the disordered N-terminus, which is not visible in the crystal structure. Q49L, H96L, and Q100L would expand the total hydrophobic area of P1 and P2.

The following figure supplements are available for figure 1:

**Figure supplement 1**. Determination of the in vivo specific activity of Spy variants.

**Figure supplement 2**. WebLogo representation of a ClustalW (**Thompson et al., 2002**) sequence alignment of 29 Spy orthologous sequences.

termination. If we succeeded at all in getting activity enhancing mutations we anticipated obtaining two types of mutations. We might obtain those that acted in a substrate specific manner that improved the action of Spy only against the substrate for which they were selected on, and variants that generally improved the activity of Spy against multiple substrates. If we succeeded in obtaining this latter type of mutations, they should be particularly informative as to what makes a protein an effective chaperone. To obtain activity-enhanced Spy variants, we used an error-prone PCR-based approach (**McCullum et al., 2010**) that targeted the mature protein encoding region of the *spy* gene on pCDFTrc-Spy to create a plasmid library of ~$10^6$ members that contained an average of 1.2 nucleotide mutations per *spy* gene. This variant library was transformed into SQ2041, a *spy* null strain of *E. coli* that contains the stability biosensor (see strain list in *Table 1*). We plated the mutant library onto LB plates that contained 0.1 mM IPTG (to induce Spy) and 4 mg/ml penicillin, the concentration at which a strain co-expressing wild-type Spy and the biosensor (strain SQ2068) fails to grow. Using this selection approach, we isolated 65 Spy variants that, when co-expressed with the biosensor, showed improved antibiotic resistance compared to cells that co-express wild-type Spy.

Remarkably, 48 (74%) of the isolated Spy variants contained a glutamine to leucine mutation at amino acid 100. For 20 of these variants, this alteration (Q100L) was the only mutation present, and strains expressing a Spy Q100L variant emerged from at least four independent mutagenesis and selection experiments. Other single mutations that answered the selection included Q25R,

**Table 1.** Strain list

| Strain | Genotype or relevant characteristics | Source |
|---|---|---|
| SQ765 | MG1655 ($F^-$ $\lambda^-$ $ilvG^-$ $rfb$-50 $rph$-1), $\Delta hsdR$ | (**Quan et al., 2011**) |
| SQ2041 | SQ765, $\Delta ampC$, $\Delta spy$, pBR322 bla::GSlinker Im7 L53A I54A (**Foit et al., 2009**) | This study |
| SQ2068 | SQ2041, pCDFTrc-Spy | This study |
| LW53 | SQ2041, pCDFTrc-Spy Q100L | This study |
| LW54 | SQ2041, pCDFTrc-Spy L32P | This study |
| LW55 | SQ2041, pCDFTrc-Spy F115I | This study |
| LW56 | SQ2041, pCDFTrc-Spy Q49L | This study |
| LW57 | SQ2041, pCDFTrc-Spy F115L | This study |
| LW58 | SQ2041, pCDFTrc-Spy H96L | This study |
| LW59 | SQ2041, pCDFTrc-Spy Q25R | This study |

L32P, and F115I. There were also a number of other mutations (Q49L, H96L, and F115L) that were found independently 2–3 times in combination with other amino acid substitutions.

To verify that these Spy mutations enhance the antibiotic resistance of the co-expressed biosensor when they are present as single mutations, we introduced the individual mutations Q25R, L32P, Q49L, H96L, Q100L, F115I, and F115L into the *spy* gene on the plasmid pCDFTrc-spy by site-directed mutagenesis and transformed the resulting plasmids into SQ2041, the *spy* knockout strain co-expressing the biosensor. All of these strains except the one containing Q49L showed improved penicillin resistance compared to strains expressing wild-type Spy at a wide range of IPTG concentrations (*Figure 1— figure supplement 1A*); the relative minimal inhibitory concentrations (MICs) were up to twofold higher than the MIC of cells co-expressing wild-type Spy.

## Improved antibiotic resistance of Spy variants is due to enhanced chaperone activity

One simple explanation for the increased penicillin resistance observed in the mutated strains might be increased Spy levels. Such an increase could occur through translational or posttranslational effects such as an increase in Spy stability. To examine these possibilities, we measured the steady-state expression levels of Spy in these strains when induced by different IPTG concentrations. All variants exhibited Spy levels that were within 20% of wild-type except Q49L, which showed a Spy level that was half that of wild-type (*Figure 1—figure supplement 1B*). These results suggest that the observed increases in MIC (up to twofold) for the variant strains are not simply due to increased expression levels of the chaperone.

We then measured the specific in vivo activity of our Spy variants by normalizing the maximal MIC values for penicillin V of the variant strains to the amount of Spy variant proteins found in these strains (*Figure 1—figure supplement 1*, 'Materials and methods'). All strains co-expressing the selected Spy variants (including Q49L) had normalized MICs 1.4–2.2-fold higher than SQ2068 (*Table 2*), indicating that the variant Spy proteins have higher specific activity than wild-type Spy. Furthermore, quantitative western blots showed that the increased MICs are linearly correlated with the steady-state levels of the folding biosensor (*Figure 1—figure supplement 1C*).

To test whether the Spy variants' increased chaperone activity was general or client specific, we purified the variant proteins and tested their chaperone activity in vitro using two standard chaperone clients: reduced denatured α-lactalbumin (α-LA) and chemically denatured aldolase. In genetic selections one usually gets what you select for, thus we had anticipated that the variants we obtained would show an improved ability to refold Im7. These mutations, would at a minimum, likely to be informative about the factors involved in Spy-Im7 interactions. We however considered it unlikely that they would show generally improved chaperone activity for at least two reasons. First, in a wide variety of laboratory evolution experiments where variant enzymes are selected that show improved activity against one substrate, often though not always show decreased activity against other unrelated substrates (*Goldsmith et al., 2012*; *Yang et al., 2013*). More specifically, other efforts at improving chaperone activity, though showing some success in generating mutants that were better with the substrates they were selected on, in general showed decreased chaperone activity against other substrates (*Wang et al., 2002*; *Aponte et al., 2010*; *Schweizer et al., 2011*). For instance Wang et al, through the use of a multistep screening process, succeeded in isolating GroEL variants that enhanced the expression of GFP and circularly permuted versions of GFP 3-8-fold, presumably by enhancing folding of GFP in vivo. However these variants were defective in all other measures of GroEL function tested including ability to support bacterial growth at high temperature, in vivo folding of the GroEL substrate HrcA, and phage lambda and Mu growth (whose growth dependency on GroEL and GroES historically led to the naming of the GroE genes [*Georgopoulos et al., 1972*]). These GroEL variants in vitro were also no better than wild type GroEL in enhancing the yield of active GFP. The authors concluded that increased GFP folding of these variants 'comes at the expense of the ability of GroEL/S to fold its natural substrates' (*Wang et al., 2002*). The majority of the DnaK variants Aponte et al isolated based on an improved ability to fold an unstable variant of chloramphenicol acetyl transferase in vivo turned out to be inferior to wild type DnaK in refolding luciferase in vitro, though it needs to be mentioned that they did succeed in isolating four variants that showed a slightly improved in vitro refolding yield for luciferase ranging from 1.2 to 1.9-fold (*Aponte et al., 2010*). Given the apparent difficulty in isolating chaperone or enzyme variants that show generally enhanced activity against a variety of substrates, we were surprised that all 7 Spy variants that we had isolated based on their ability to fold Im7 in vivo were

**Table 2.** Properties of Spy variants

| Spy variants | MIC$_{norm}$ | Activity (aldolase agg. Prev) | Activity (aldolase refold) | Activity (α-LA agg. Prev) | k$_{on}$ (× 10$^5$ mol$^{-1}$ s$^{-1}$) | k$_{off}$ (s$^{-1}$) | KD (μM) | Tm (°C) | ΔHm (Kcal mol$^{-1}$) | ΔCp (Kcal K$^{-1}$ mol$^{-1}$) | ΔG$_{NU}$ (25°C) (Kcal mol$^{-1}$) |
|---|---|---|---|---|---|---|---|---|---|---|---|
| WT | 1 | 1 | 1 | 1 | 3.98 ± 0.11 | 0.456 ± 0.011 | 1.15 ± 0.027 | 48.1 ± 0.1 | 66.6 ± 1.5 | 0.64 | 4.24 ± 0.10 |
| Q25R | 1.44 | 6.90 ± 0.64 | 1.38 ± 0.31 | 2.44 ± 0.89 | 2.29 ± 0.13 | 0.198 ± 0.005 | 0.87 ± 0.034 | 46.3 ± 0.4 | 73.7 ± 0.8 | 0.98 | 4.20 ± 0.10 |
| L32P | 1.92 | 2.52 ± 0.11 | 4.85 ± 0.57 | 2.10 ± 0.74 | 1.51 ± 0.14 | 0.030 ± 0.002 | 0.20 ± 0.003 | 31.0 ± 0.2 | 52.1 ± 2.5 | 0.71 | 0.99 ± 0.02 |
| Q49L | 1.60 | 2.88 ± 0.14 | 4.25 ± 0.66 | 1.93 ± 0.66 | 2.30 ± 0.06 | 0.176 ± 0.009 | 0.76 ± 0.018 | 52.0 ± 0.2 | 59.9 ± 1.0 | 0.68 | 4.19 ± 0.10 |
| H96L | 1.62 | 2.02 ± 0.32 | 1.90 ± 0.32 | 1.64 ± 0.49 | 2.68 ± 0.14 | 0.266 ± 0.010 | 1.00 ± 0.022 | 50.1 ± 0.1 | 56.2 ± 3.0 | 0.71 | 3.66 ± 0.23 |
| Q100L | 2.19 | 1.34 ± 0.05 | 4.20 ± 0.44 | 2.12 ± 0.75 | 1.19 ± 0.13 | 0.027 ± 0.002 | 0.23 ± 0.013 | 53.8 ± 0.6 | 28.9 ± 1.1 | 0.23 | 2.24 ± 0.11 |
| F115L | 1.52 | 1.98 ± 0.32 | 4.85 ± 0.56 | 2.30 ± 0.83 | 2.73 ± 0.01 | 0.245 ± 0.007 | 0.90 ± 0.028 | 41.3 ± 0.2 | 56.6 ± 4.9 | 0.76 | 2.60 ± 0.22 |
| F115I | 1.65 | 2.21 ± 0.12 | 4.33 ± 0.49 | 2.34 ± 0.85 | 2.82 ± 0.15 | 0.328 ± 0.017 | 1.17 ± 0.097 | 41.7 ± 0.4 | 54.3 ± 1.5 | 0.98 | 2.43 ± 0.11 |

ΔG$_{NU}$(25°C) is the free energy of stabilization at 25°C (NU dictates the transition from folded state to unfolded state), ΔH$_m$ is the change in enthalpy at T$_m$ which is the melting temperature and ΔCp is the change in heat capacity associated with the unfolding of the Spy variant. agg. prev: aggregation prevention. Fold activity expresses relative to WT. Values after the ± sign are standard errors.

MIC$_{norm}$ is measured for cells (SQ2068, LW53-59) expressing the pBR322 bla::GSlinker Im7 L53A I54A plasmid and various Spy constructs. k$_{on}$, k$_{off}$, and KD are kinetic parameters describing the interaction between Im7 L53A I54A and the Spy variants.

significantly more active in preventing aggregation of both chemically denatured α-lactalbumin and aldolase in in vitro assays. In the aldolase aggregation assay, they were 1.3–6.9-fold more active than wild type, and in the α-lactalbumin aggregation assay, they were 1.6–2.4-fold more active (*Table 2*). We also tested for the activity of these chaperone variants in their ability to facilitate aldolase refolding. Six of the seven of the variants were found to be more active than is wild type Spy in the range of 1.9–4.9-fold. The one exception, Q25R, was measured to marginally increase refolding yield (1.4-fold) (See *Table 2*). Because our Spy variants showed improved chaperone activity towards at least three client proteins (Im7, aldolase and α-lactalbumin), we called them 'super-Spy' variants.

## Hydrophobic areas may be involved in client binding of activity-enhanced Spy variants

We mapped the activity-enhancing mutations identified in the selection onto Spy's crystal structure and found that many of them were located close to each other. Most of them mapped immediately adjacent to the two predominant hydrophobic patches on the concave surface (P1, P2) of the cradle interior (*Figure 1A,B*)—the region that we had previously hypothesized might be involved in client binding (*Quan et al., 2011*). The most commonly observed substitutions (Q100L, which occurred in 74% of the variants, and H96L and the Q49L, which occurred in ~5% and ~3% of the variants, respectively) change polar or charged residues (glutamine and histidine) into the hydrophobic amino acid leucine, thereby increasing the area of the hydrophobic region.

One proposed mechanism for chaperone function is via blocking hydrophobic regions present on client proteins, thereby preventing their aggregation (*Hartl et al., 2011*). Simple expansion of peptide-binding hydrophobic regions on our Spy variants could thus be one straightforward way to explain their improved chaperone activity. There are some expectations of this simple model:(1) chaperone variants with a larger or stronger hydrophobic patch will have enhanced affinity for client proteins, and (2) client proteins are likely to interact with regions on the 3D structure of the chaperone that are adjacent to the sites mutated in our selected Spy variants. Alternatively, the Spy mutations could increase chaperone efficacy in other less direct ways. For example, they could map to sites distant from the active site of the chaperone and exert their beneficial action through allosteric effects. To help distinguish between these possible models and to better understand how Spy interacts with its clients, we decided to map the site(s) with which Spy binds the client protein Im7. To achieve this, we:(1) examined the effects of Im7 binding on hydrogen-deuterium exchange in Spy, (2) investigated the proteolytic sensitivity of the chaperone in the presence and absence of Im7, and (3) crosslinked Spy to its client.

## Hydrogen-deuterium exchange identifies Spy residues involved in client binding

Hydrogen-deuterium exchange at individual peptide bond amides is determined by the protection of amides from solvent, either due to maintenance of secondary and/or tertiary structure or client binding. To ascertain the effects of client protein binding on deuterium exchange, we compared the level of hydrogen-deuterium exchange of free Spy with that of a Spy-Im7 complex using a mass spectrometry approach. We incubated Spy with a 40-amino acid peptide derived from Im7 L53A I54A (residues 7–45), which binds Spy with a 2.6 μM KD (see 'Materials and methods' for details of peptide generation and selection). Using 10s exchange times, we obtained evidence that ~10 Spy amides become more protected upon Spy-Im7 complex formation compared to free Spy (*Figure 2—figure supplement 1A*). Substantial changes in protection were mapped to the residues located in N- and C-terminal regions of Spy (*Figure 2—figure supplement 1B*). We were able to localize the improved Spy protection in the presence of client to several specific residues: T5, H16, A37, Q114, F115, F119, and E125. The protection includes not only the flexible N and C termini (residues 1–28 and 125–138), which are not present in the crystal structure, but extends into the α1 and α4 helices as well (*Figure 2*). Increased protection from hydrogen-deuterium exchange in these regions could possibly indicate involvement of these Spy residues in the interaction with Im7 or could imply the folding of these flexible regions upon client binding, or a combination of both.

## Limited proteolysis reveals potential Im7 binding sites in Spy

As a complementary approach to probe the Im7 binding site in Spy, we used a limited proteolysis assay with a mass spectrometric readout to characterize the exposed and buried regions in Spy before and after Im7 binding as suggested by their accessibility to the protease trypsin. Proteolytic sites in

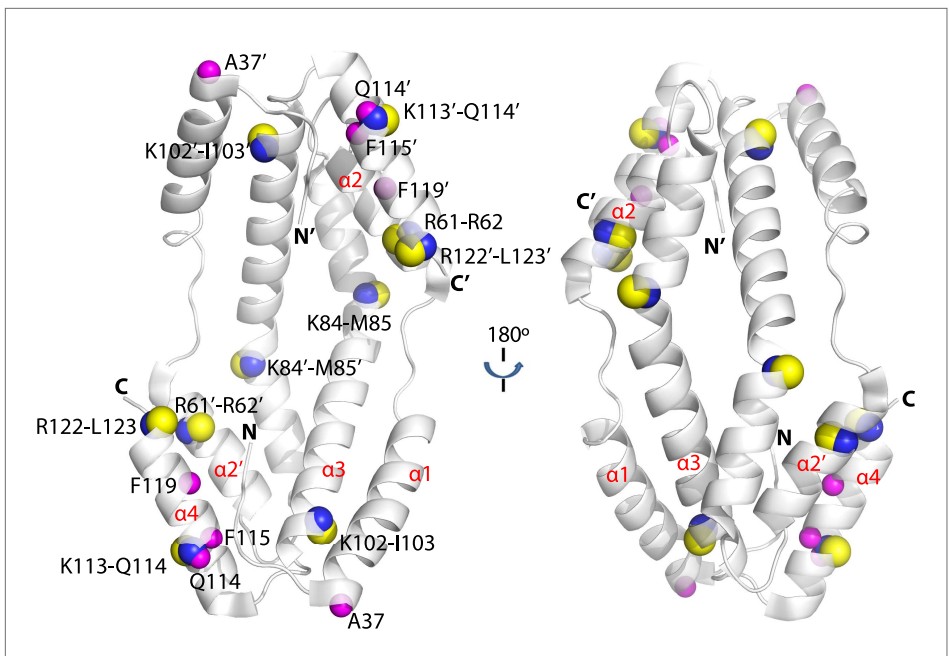

**Figure 2**. Hydogen-deuterium exchange and limited proteolysis reveal potential hot spots on Spy for Im7 L53A I54A or Im7 7-45 peptide binding. Hydrogen atoms on the backbone amide bond that are protected upon addition of Im7 7-45 are shown as magenta spheres. Peptide bonds of Spy protected from trypsin upon addition of Im7 L53A I54A are shown as yellow and blue spheres, with yellow representing carbon atoms and blue representing nitrogen atoms.

The following figure supplements are available for figure 2:

**Figure supplement 1**. Hydrogen-deuterium exchange analysis of Spy and the Spy-Im7 7-45 complex by electron capture dissociation fourier transform ion cyclotron resonance mass spectrometry (ECD-FTICR-MS).

**Figure supplement 2**. Limited proteolysis reveals potential Im7 binding sites in Spy.

Spy that show altered trypsin susceptibility may either be directly involved in Im7 binding or be near the Im7 binding site. Note that it is also possible that Im7 binding induces a significant conformational change or change in flexibility in Spy that alters the trypsin susceptibility of certain residues. In the absence of Im7, the flexible N terminus and to a lesser extent the flexible C terminus of Spy are more accessible to trypsin than the structured regions (*Figure 2—figure supplement 2*). In the presence of Im7, a number of sites including R61, K84, K102, K113, and R122 show significant protection compared to free Spy (*Figure 2*). Notably, changes in trypsin susceptibility occur on both the inside and outside of the cradle due to the thinness of the Spy molecule. These susceptibility changes suggest that either client binding occurs over large portions of Spy or that client binding involves major conformational changes or changes in flexibility, or a combination of these factors. Combining these observations with the deuterium exchange results suggests that Im7 peptide binding affects a relative large area on the Spy surface, especially the rim regions (α4 helix and the N-terminus of the α2' helix) and the tips (N-terminus of the α1 helix, C-terminus of the α3 helix, and the N-terminus of the α4 helix) of the Spy cradle (*Figure 2*).

## Crosslinking reveals key residues directly involved in interactions with client

To further map the position of the client-binding site on Spy, we performed crosslinking analysis. Crosslinking provides information about the distances between two cross-linked residues as determined by the length of the spacer in the crosslinking reagent. Identification of the crosslinked sites on a protein complex thus provides spatial information and distance constraints for the two amino acid residues that are crosslinked.

We performed crosslinking experiments on Spy and the peptide composed of residues 7-45 from Im7. Crosslinking was done using our recently developed isotopically-coded collision-induced dissociation CID-cleavable affinity-purifiable amine-reactive 14 Å length crosslinker CyanurBiotinDimercaptoPropionylSuccinimide (CBDPS-H8/D8) (*Petrotchenko et al., 2011*) and the newly developed azidobenzoicacidsuccinimide (ABAS-$^{12}$C6/$^{13}$C6), an isotopically-coded photo-reactive 7 Å length crosslinker. Given the relatively long span length of the CBDPS crosslinkers and the flexibility of the crosslinked side chains, we were not surprised to find multiple CBDPS Lys–Lys crosslinks (*Figure 3*; *Table 3*). The most frequently crosslinked residues are K18 and K20 on the unstructured N terminus of Spy. T5 and H16 in this region are also implicated in peptide binding through changes in deuterium protection, suggesting that this flexible N terminus of Spy might be involved in client interaction.

The shorter the crosslinking reagent, the more precise and definitive the structural information that crosslinking analysis can provide. To obtain such short-distance constraints for the Spy-Im7 complex, we additionally performed crosslinking using two zero-length crosslinking reactions: tyrosine reactive, Photo-Induced Cross-Linking of Unmodified Proteins (PICUP) (*Bitan et al., 2001*) and carboxyl/amine reactive 1-ethyl-3-(3-dimethylaminopropyl)carbodiimide (EDC). Using these zero-length reagents, we were able to confidently detect and identify additional short distance Spy-Im7 crosslinks between Y104 of Spy and Y10 of Im7, and between K39 of Spy and E12 of Im7 (*Table 3*; *Figure 4*). Notably, nearly all crosslinks identified in all four crosslinking reactions occurred on the concave side of Spy, with a few occurring on the rim of the cradle (*Figure 3*). Essentially no crosslinks were obtained on the convex side of Spy despite the abundance of lysines on the convex side (*Figure 3—figure supplement 1*), providing evidence that peptide binding occurs on the interior concave surface of the Spy homodimer (*Figure 3*). The zero-length PICUP crosslinks to Y104 independently suggests that the concave surface is the interface at which Im7 binds.

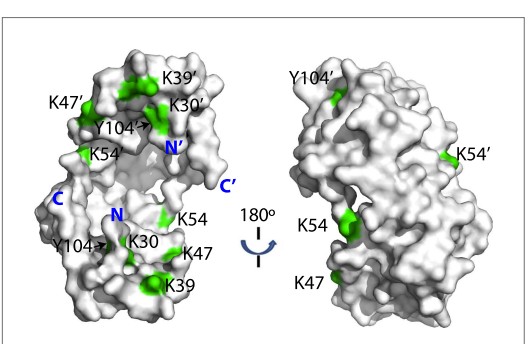

**Figure 3**. Crosslinked residues mapped onto the crystal structure of Spy. Spy residues that were found crosslinked to Im7 7-45 peptide are shown in green. Crosslinking with CBDPS-H8/D8 implies a short distance between the N terminus of Im7 and Spy residues; these include Spy K20, K39, K47, K54, K130, and K132. Residues that can be crosslinked with CBDPS also include Spy K18 to Im7 K20, Spy K20 to Im7 K20, and Spy K30 to Im7 K43. Using ABAS, we identified crosslinking between the N terminus of Spy and E21 of Im7. Zero-length crosslinking using PICUP and EDC reagents identified contacts between Spy Y104 and Im7 Y10, and Spy K39 and Im7 E12, respectively. A summary of all identified crosslinks is provided in *Table 3*.

The following figure supplements are available for figure 3:

**Figure supplement 1**. Distribution of lysine residues on the surface of Spy, lysine residues are copper colored.

## Modeling of the Spy and Im7 7-45 complex

Using a hierarchical approach that consists of three steps (docking pose generation, decoy clustering, and structure refinement) (see 'Materials and methods'), we built a tentative, theoretical model of the Spy-Im7 complex by docking the Spy structure (PDB ID: 3O39 ) with the Im7 7-45 peptide structure (modeled using I-TASSER [*Zhang, 2008*; *Roy et al., 2010*]). Note that the distance constraints we obtained from the crosslinking study were NOT applied during the docking and refinement process. However, of the top 10 models with the lowest energy scores, six fit the zero length crosslinker data (i.e., the critical constraints for the two pairs of residues that crosslinked with the zero length crosslinkers: Spy Y104 and Im7 Y10, Spy K39 and Im7 E12), with atomic contact distances ranging between 2.5 and 6.5 Å. These models are very similar with average pair-wise RMSDs of about 2 Å. Thus, the docking analysis provides evidence that the crosslinking experiments were sampling an energetically plausible complex. For further analysis, we selected the model with the smallest atomic contact distance for these two pairs of residues found to interact by zero-length crosslinkers.

Im7 7-45 was modeled to adopt a two-helix hairpin conformation and to fit into the concave face of the Spy dimer (*Figure 4*), consistent with the $^{15}$N HSQC NMR spectrum of $^{15}$N-labeled Im7

**Table 3.** Spy-Im7 crosslinks

| Mass (Da) | Rt (min) | Δ (ppm) | Pr 1 | S | E | Res | | Sequence* | | Pr2 | S | E | Res | | Sequence* | | CL | Enz |
|---|---|---|---|---|---|---|---|---|---|---|---|---|---|---|---|---|---|---|
| 1853.87609 | 16.47 | 0.4 | Spy | - | 12 | - | - | **S**ADTTTAAPADAK† | P | Im7 | 21 | 24 | 21 | K | **E**IEK | E | ABAS | Tr |
| 1835.77969 | 20.77 | 0.6 | Spy | 15 | 18 | 18 | M | MHH**K** | G | Im7 | 20 | 25 | 20 | L | **K**EIEKE | N | CBDPS | PK |
| 1447.60195 | 22.43 | 0.6 | Spy | 16 | 18 | 18 | M | HH**K** | G | Im7 | 20 | 23 | 20 | L | **K**EIE | K | CBDPS | PK |
| 1704.73954 | 19.18 | 0.5 | Spy | 16 | 18 | 18 | M | HH**K** | G | Im7 | 20 | 25 | 20 | L | **K**EIEKE | N | CBDPS | PK |
| 2328.05587 | 21.48 | 1 | Spy | 16 | 24 | 20 | M | HH**K**GKFGPH | Q | Im7 | 20 | 25 | 20 | L | **K**EIEKE | N | CBDPS | PK |
| 1567.68178 | 21.40 | −0.2 | Spy | 17 | 18 | 18 | H | H**K** | G | Im7 | 20 | 25 | 20 | L | **K**EIEKE | N | CBDPS | PK |
| 3673.64206 | 53.63 | −0.4 | Spy | 19 | 30 | 20 | K | G**K**FGPHQDMMFK | D | Im7 | - | 20 | - | - | **S**ISDYTEAEFVQLLK | E | CBDPS | Tr |
| 1781.79357 | 34.95 | −0.9 | Spy | 19 | 24 | 20 | K | G**K**FGPH | Q | Im7 | 19 | 23 | 20 | L | L**K**EIE | K | CBDPS | PK |
| 2038.92970 | 31.40 | −0.1 | Spy | 19 | 24 | 20 | K | G**K**FGPH | Q | Im7 | 19 | 25 | 20 | L | L**K**EIEKE | N | CBDPS | PK |
| 1925.84667 | 28.32 | −0.6 | Spy | 19 | 24 | 20 | K | G**K**FGPH | Q | Im7 | 20 | 25 | 20 | L | **K**EIEKE | N | CBDPS | PK |
| 1377.61083 | 33.10 | 0.3 | Spy | 29 | 31 | 30 | M | F**K**D | L | Im7 | 42 | 45 | 43 | F | V**K**IT | - | CBDPS | PK |
| 3793.80025 | 56.15 | −0.5 | Spy | 31 | 43 | 39 | K | DLNLTDAQ**K**QQIR | E | Im7 | - | 20 | - | - | **S**ISDYTEAEFVQLLK† | E | CBDPS | Tr |
| 3266.68723 | 45.13 | 1.1 | Spy | 31 | 43 | 39 | K | DLNLTDAQ**K**QQIR | E | Im7 | - | 20 | 12 | - | SISDYT**E**AEFVQLLK | E | EDC | Tr |
| 3112.43551 | 53.53 | 0.1 | Spy | 44 | 50 | 47 | R | EIM**K**GQR | D | Im7 | - | 20 | - | - | **S**ISDYTEAEFVQLLK | E | CBDPS | Tr |
| 3649.69333 | 51.73 | −0.7 | Spy | 51 | 61 | 54 | R | DQM**K**RPPLEER | R | Im7 | - | 20 | - | - | **S**ISDYTEAEFVQLLK | E | CBDPS | Tr |
| 1838.84488 | 31.50 | 0.5 | Spy | 54 | 61 | 54 | M | **K**RPPLEER | R | Im7 | - | 8 | - | - | **S**IS | D | CBDPS | PK |
| 2958.53428 | 48.48 | 0.1 | Spy | 103 | 112 | 104 | K | I**Y**NILTPEQK | K | Im7 | - | 20 | 10 | - | SISD**Y**TEAEFVQLLK | E | PICUP | Tr |
| 1699.77183 | 31.88 | −0.3 | Spy | 123 | 130 | 130 | R | LTERPAA**K** | G | Im7 | - | 8 | - | - | **S**IS | D | CBDPS | PK |
| 2822.33254 | 55.33 | −0.5 | Spy | 127 | 132 | 130 | R | PAA**K**GK | M | Im7 | - | 20 | - | - | **S**ISDYTEAEFVQLLK | E | CBDPS | Tr |
| 3055.37181 | 57.27 | −1.6 | Spy | 131 | 138 | 132 | K | G**K**MPATAE | - | Im7 | - | 20 | - | - | **S**ISDYTEAEFVQLLK | E | CBDPS | Tr |

Rt: retention time; Δ: mass error for crosslink assignments; Pr: protein; S, E: starting and ending amino acid residues (sequence numbers) of the crosslinked peptides, respectively; Res: crosslinked residue (sequence number) within corresponding peptide; CL: crosslinking reagent used; Enz: digestion enzyme used. The crosslinked residues are bolded and underlined in the sequences.

*The residues shown before and after the sequences are the preceding and following residues of the peptide sequence. They are shown to illustrate digest specificity.

†The N-terminal serine was introduced from the Sumo fusion constructs, which were used for Spy or Im7 purification (**Quan et al., 2011**). The symbol '−' is used to indicate this N-terminal serine.

7-45, which suggests that although it is largely unfolded in solution, it is perhaps partially biased towards α-helix formation (**Figure 4—figure supplement 1**). Given that this model is strongly constrained by only two pairs of residues that were found to interact using the zero-length crosslinkers, it is best regarded as a very tentative and theoretical model. Never-the-less it is consistent with our experimental data and helps in our interpretation of it. The contacts predicted between Im7 and Spy in this tentative model are moderately extensive, burying a surface of ~1502 Å², which is consistent with medium binding affinity (KD = 2.6 µM) (**Chen et al., 2013**). The interactions between Im7 and Spy in the model are mediated by a combination of electrostatic and hydrophobic interactions. The two helices of Im7 contain a number of acidic and polar residues that dock into the positive-charged concave surface of Spy (**Figure 4—figure supplement 2A–C**). The α1 helix of Im7 is surrounded primarily by basic residues from one Spy monomer, whereas α2 of Im7 is symmetrically coordinated by the basic patch from the other Spy monomer.

The N-terminal loop of Im7 forms extensive van der Waals contacts with a cluster of hydrophobic residues at the tip of the hydrophobic patch P1 of Spy in our tentative model. The most prominent residue is Im7 Y10, which is buried into a hydrophobic pocket surrounded by L32, L34, I42, M46, I103, Y104, L107, and F115 of Spy (**Figure 4—figure supplement 2D**). Satisfyingly, these residues include those that make up the P1 patch shown in **Figure 1** and some of the hydrophobic residues at the tip of P1 (**Quan et al., 2011**). The distance between Im7 Y10 and Spy Y104 is 4.5 Å in this model, consistent with the close distance constraint that was defined by the zero-length crosslinking via PICUP. Similarly,

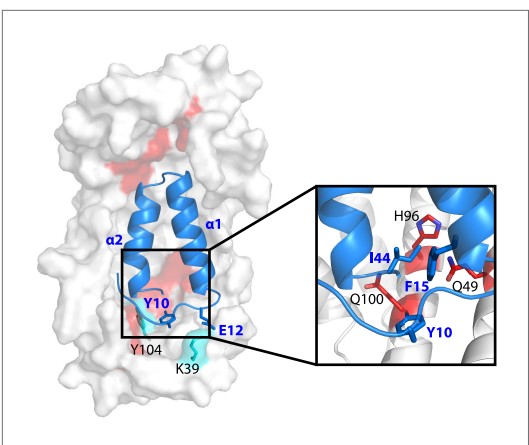

**Figure 4**. Model showing the position of the Im7 7-45 peptide bound on the concave surface of Spy. Mutations that increase the specific activity of Spy are shown in red. Residues on Spy that are crosslinked to Im7 residues by PICUP or EDC are shown in cyan. The enlargement shows the position of residues Q49, H96, and Q100. Mutating these residues to leucine increases the hydrophobic interaction with Y10, F15, and I44 on Im7.

The following figure supplements are available for figure 4:

**Figure supplement 1**. NMR spectrum of the Im7 7-45 peptide.

**Figure supplement 2**. Models of the Spy-Im7 7-45 complex.

Im7 Glu12 and Spy Lys39 form a direct salt bridge, consistent with their interaction via the zero-length crosslinker EDC. This model also helps explain the effects of at least some of our beneficial Spy mutations. Polar-to-apolar Spy mutations (Q49L, H96L, Q100L) may enhance the hydrophobic interaction that Spy has with Im7 through the interaction with Y10, F15, and I44 of Im7 by expanding and partially fusing the hydrophobic patches P1 and P2 on Spy (*Figures 1 and 4*).

## Super-Spy variants release client proteins more slowly in vitro

To further understand why these super-Spy variants enhance the expression and presumably the in vivo stability of Im7 L53A I54A, we measured their interaction affinities and kinetics with Im7 L53A I54A using bio-layer interferometry (BLI). Biotinylated Im7 L53A I54A was immobilized on the streptavidin coated sensor tip. Binding of Spy to Im7 alters the thickness of the molecular layer on the tip surface, which triggers a change in the spectrum signal (*Abdiche et al., 2008*). Thus, we can monitor the binding in real time to measure the association and dissociation rates of the two proteins. In order to get accurate $k_{on}$ and $k_{off}$ rates we found it is vital to use a substrate that is soluble and not bound to the tip as an aggregate. Unfortunately, most chaperone substrates such as aldolase rapidly aggregate when placed under conditions where they are chaperone substrates (i.e., at least partially unfolded) making it very difficult to accurately determine $k_{on}$ and $k_{off}$ rates. Im7 L53A I54A has the very fortunate properties of being not only a clear in vivo substrate of Spy (indeed Spy was discovered by the ability it has to enhance the yield of folded Im7 L53A I54A) but also soluble in solution. This nicely behaved, soluble chaperone substrate bound to the tips in a reproducible manner and allowed us to determine the $k_{on}$ and $k_{off}$ rates for the chaperone. All of the Spy variants we tested showed smaller values for both $k_{on}$ and $k_{off}$ compared to wild-type Spy (*Table 2*), suggesting that they both bind and release clients more slowly. Overall, the decreases in the $k_{off}$ rates are more dramatic than the decreases in the $k_{on}$ rates (for instance, Q100L shows a ~three-fold decrease in $k_{on}$ and a ~16-fold decrease in $k_{off}$). As a result, all selected Spy variants except F115I show a significantly increased apparent affinity for Im7 L53A I54A, up to 5.8 -fold. Note that Q100L, the mutation present in 74% of the variants that answered our selection, has the largest effect on both $k_{on}$ and $k_{off}$ of any of the variants tested. It is thus a good possibility that the increase in the ability of at least some of our Spy variants to stabilize Im7 in vivo is at least in part due to their increased affinity for Im7.

## Super-Spy variants are thermodynamically less stable and likely more flexible than wild-type Spy

Spy is a very thin molecule, almost entirely lacking a hydrophobic core. This unusually thin nature of Spy is presumably under genetic selection and of functional significance. It also may allow for conformational changes during Spy's chaperone cycle. Consistent with this hypothesis, major changes in the fluorescence of environmentally sensitive probes attached to various locations in Spy were observed to occur upon client protein binding (*Quan et al., 2011*). One possibility is that our mutations act to alter the flexibility and stability of Spy and, in so doing, alter its chaperone activity.

To investigate if the mutations in Spy resulted in changes in the protein's stability, we performed thermal denaturation experiments to measure the free energy of unfolding. Five of the seven super-Spy

variants were significantly less thermodynamically stable than the wild-type protein (*Table 2*), with mutant L32P being the most destabilized ($\Delta\Delta G_{NU}$ = −3.25 kcal mol$^{-1}$). The Q25R and Q49L variants had stabilities that were indistinguishable from wild type. When we plotted thermodynamic stability vs in vivo chaperone activity, we found a significant ($R^2$ = 0.46) negative correlation (*Figure 5A*). Thus the *less* thermodynamically stable the Spy protein is, the *better* it tends to function as a chaperone in vivo. There is also a significant correlation ($R^2$ = 0.51) between the thermodynamic stability of these Spy variants and their KD of binding Im7 L53A I54A, with the least stable variants showing the tightest binding (*Figure 5B*). One possibility is that decreased stability results in increased flexibility, which may then result in improved chaperone activity by allowing for more adaptive and tighter binding to client proteins.

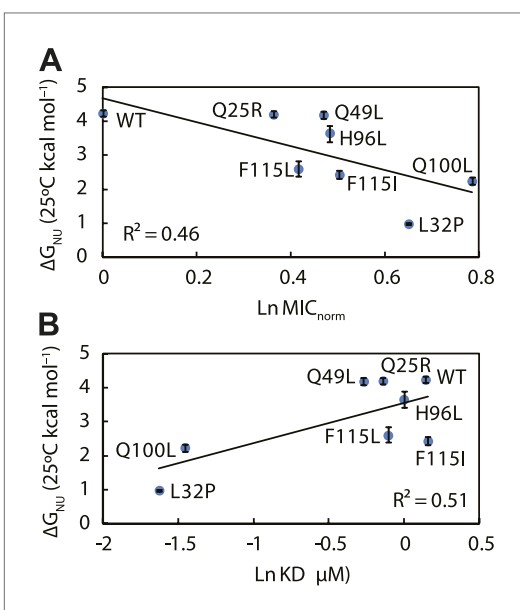

**Figure 5**. The thermodynamic stability of the Spy variants is inversely correlated with their chaperone activity and their affinity for its client protein. (**A**) The in vivo chaperone activity of Spy variants is expressed as the normalized relative MIC of the strains expressing the variants plus the Bla-Im7 L53A I54A biosensor (SQ2068, LW53-59). (**B**) The binding activity of Spy variants towards Im7 L53A I54A is expressed as their dissociation constant (KD) to the client; smaller values indicate tighter binding.

The following figure supplements are available for figure 5:

**Figure supplement 1**. The more thermodynamically stable Spy variants are less flexible.

**Figure supplement 2**. Calculation of relative Spy activity in preventing α–lactalbumin (α -LA) aggregation.

**Figure supplement 3**. Calculation of relative Spy activity in preventing aldolase aggregation.

**Figure supplement 4**. Calculation of the relative activity of Spy variants in the refolding of denatured aldolase.

As another measure of apparent flexibility, we performed deuterium exchange analysis on these Spy variants in the presence and absence of client. Without the Im7 7-45 peptide, the apparent flexibility of Spy variants as measured by the percentage of protected protons inversely correlates with their thermodynamic stability measured at the same temperature (25°C) (*Figure 5—figure supplement 1*). Note that several of the super-Spy chaperones including L32P, F115L, and F115I show either zero or a very small number of protected protons (0, 0, and 5, respectively), implying a very high degree of disorder in the absence of client proteins. On the other hand, all of our Spy variants show almost the same level of protection when the Im7 7-45 peptide is added, suggesting either that they achieve a similar level of order upon interaction with the client, or that the vast majority of Spy is directly protected via contact with Im7. These results further imply that binding to the client may induce the folding of the very unstable Spy variants such as L32P.

## Discussion

Although molecular chaperones assist in the folding of a vast number of cellular proteins, including many that are linked to disease states, the mechanism by which they accomplish this feat is still not completely clear (*Horwich et al., 2009*; *Kalia et al., 2010*; *Hartl et al., 2011*). Defining the binding interface between a chaperone and a client would undoubtedly be very valuable in understanding chaperone action, but these binding sites remain poorly defined. Various regions on the small heat shock proteins, for instance, have been implicated in client binding, including both the N and C termini and sites within the core-alpha crystalline domain (*Jaya et al., 2009*; *Basha et al., 2013*). Unfortunately, we only have a few structures of chaperone-client complexes (*Martinez-Hackert and Hendrickson, 2009*; *Zhu et al., 1996*; *Zahn et al., 2013*; *Bracher et al., 2011*) and importantly, these do not resolve how a chaperone interacts throughout its cycle.

To better understand how chaperones bind to proteins, we decided to take a genetic approach

with the aim of improving the ability of a chaperone called Spy to protect a poorly folding protein from degradation in vivo. In our selection, we obtained several Spy variants that appear to act as improved chaperones by expanding and partially fusing two hydrophobic patches present on the interior of the cradle-like structure of Spy. By using crosslinking, proteolytic sensitivity, and deuterium protection experiments, we provide additional evidence that implicates these regions in client binding. Our super-Spy variants demonstrate tighter client binding, and we observe a correlation between the in vitro dissociation constants of these variants to the client protein Im7 L53A I54A and their ability to stabilize Im7 L53A I54A in vivo (*Figure 6*). This implies that Spy's binding affinity for Im7 L53A I54A is important in determining its chaperone activity. Tighter binding could act to decrease the steady state concentration of presumably aggregation-prone or protease sensitive Im7 folding intermediates, thus increasing the Im7 level in the cell. Variants of Spy that increased the affinity of clients beyond a certain point may not be obtained by our in vivo selection because they would be expected to fail to release their clients in a timely fashion. Further increases in affinity are expected to be counterproductive. For aldolase aggregation inhibition Q25R, for instance, is the most effective, but it is only very marginally better than wild type Spy in refolding aldolase, perhaps partly because it fails to release aldolase at an optimal rate. Client and variant specific changes in affinity may explain why there appears to be no good relationship between the activity that one specific variant shows with the various specific client proteins tested.

The genetic selection we performed was specifically designed to enhance the ability of Spy to stabilize a particular mutant of Im7 (Im7 L53A I54A). This selection procedure was expected to result in identifying client-optimized Spy variants specialized for the unstable Im7 L53A I54A but with lower levels of chaperone activity on other clients. A previous attempt to evolve the chaperone GroEL for instance, succeeded in generating highly client specific variants that were defective in the folding of other clients (*Wang et al., 2002*). We were therefore surprised, but pleased, to see that all the Spy variants appeared to function more efficiently than wild-type Spy in preventing aggregation of aldolase and α-lactalbumin in vitro.

Based on our biochemical and biophysical evaluation of the super-Spy variants, there appear to be multiple factors that result in the increased chaperone activity. In addition to increasing binding site hydrophobicity, most of our super-Spy variants decrease Spy's thermodynamic stability and increase the level of disorder that Spy shows in the absence of client proteins. Two of the Spy variants, L32P and F115L, show no protected amide protons whatsoever in the absence of the client, implying extreme levels of flexibility prior to client binding. These data suggest that Spy flexibility is important for chaperone function, presumably by facilitating client binding. The six long coiled-coil tentacles in the jellyfish-like chaperone prefoldin, for instance, are flexible, enabling it to adjust its central cavity and capture various client proteins (*Siegert et al., 2000*). Many other chaperones including GroEL, DnaK, and the small HSPs contain regions of disorder, which have been proposed to aid in client recognition and chaperone function, although the exact roles of these disordered regions are unclear (reviewed in *Bardwell and Jakob (2012)*). There are a number of chaperones that are conditionally disordered and that are active as a chaperone in the disordered state. These include HdeA, a chaperone that is activated on the protein level by acidic pH, and Hsp33, a chaperone that

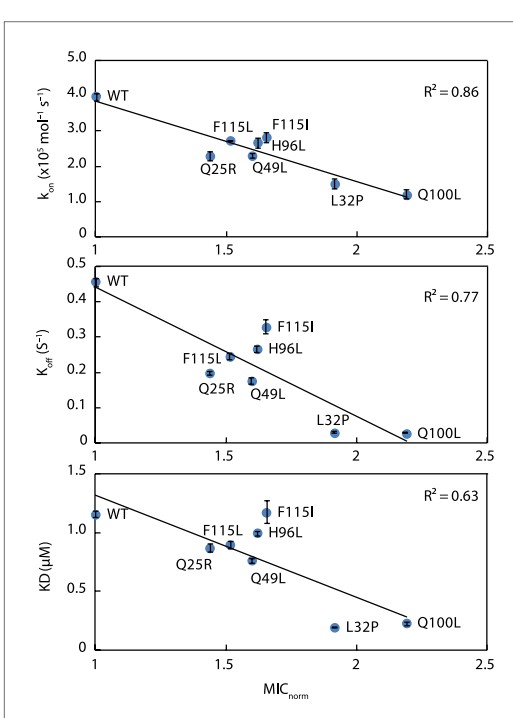

**Figure 6**. Kinetic parameters characterizing the interaction of Spy variants with the client protein Im7 L53A I54A. A linear correlation is seen between these in vitro parameters and the in vivo activity of these variants towards the same client (expressed as normalized MIC). Spy variants with better activity in vivo bind and release client slower and have an overall tighter affinity for the client.

is activated by oxidation (*Bardwell and Jakob, 2012*; *Foit et al., 2013*). Disorder is also an emerging theme, not only among chaperones but among many hub proteins, which are capable of binding to multiple unrelated targets (*Bardwell and Jakob, 2012*). It has been proposed that proteins utilize these disordered regions in the adaptive recognition of their various binding partners (*Oldfield et al., 2008*); yet, apart from a few exceptions (*Brzovic et al., 2011*), defining the precise roles that disorder plays in these promiscuous binding proteins has proved to be difficult.

In general, the majority of mutations in proteins are destabilizing (*Foit et al., 2009*), so if destabilization was the only requirement for enhanced chaperone ability, a wider range of mutations might be expected to destabilize Spy and thus increase its flexibility. If the only requirement is to increase the hydrophobicity of the P1 and P2 region, one might also expect a larger variety of mutants to have answered the genetic selection. The very narrow range of substitutions that answers our selection with 74% of our variants containing the Q100L substitution strongly suggests that these variants may be having more specific effects beyond their effects on surface hydrophobicity and flexibility. Several of the substitutions do not appear to result in a substantial gain in surface hydrophobicity. One alteration, F115L/or I, replaces one hydrophobic residue with another, and two of the substitutions, Q25R and L32P, apparently result in a decrease in hydrophobicity, though not specifically for the patches P1 and P2. Despite these exceptions, it is striking that the majority of the variants isolated do exhibit changes in flexibility and surface hydrophobicity, which goes at least part way toward explaining their 'super-Spy' activity.

If the variants we observed do generally enhance Spy's chaperone activity, then why has evolution not already come up with this solution? In short, it has. Inspection of an alignment of Spy homologues (*Figure 1—figure supplement 2*) shows that although evolution has, for some of the residues commonly made the exact same substitutions as have emerged from our selection, and in other cases it has found very similar answers. The F115L substitution is very commonly found in Spy homologue, as is Q49L. The precise substitution that we found, H96L has not been observed in the sequences we compared, but remarkably, the chemically very similar residue M is actually the most common residue found at position 96. Evolution generally acts to optimize rather than maximize protein activity. One can imagine that some organisms require more or less Spy activity. The optimal level of Spy activity selected for in *E. coli* is apparently less than seems to be required in other organisms. That other organisms have independently obtained the same 'Super Spy' substitutions that we found is testimony of the power of our selection for in vivo stability.

## Materials and methods

### Spy library construction and selection

Plasmid pCDFTrc-spy was used as the template for the construction of the Spy mutant library in an error-prone PCR reaction (*McCullum et al., 2010*). In order to introduce random mutations only into the mature protein coding region of Spy, we first introduced a BglI site between the signal sequence and the mature protein coding region of Spy and then amplified only this region using the forward primer containing the BglI site (5'CGC GGC CAA CCT GGC CCA TGC C3') and the reverse primer bearing the EcoRV site that is located at the 3' end of the Spy gene (5'GGC CGA TAT CCA ATT GAG ATC TGC CAT ATG GGA TCC TTA3'). The linear fragments were digested with BglI and EcoRV and then re-ligated into the pCDFTrc vector (digested with BglI and EcoRV as well). We used four different concentrations of template DNA (0.05, 0.5, 5, and 50 ng) to obtain a broad spectrum of mutation frequencies as the lower concentrations of the template are expected to give rise to higher mutation rates. The mutation rates were 1.8, 5.0, 5.3, and 4.3 mutations per 1000 nucleotides using 50, 5, 0.5, and 0.05 ng of template DNA, respectively. Sequencing of 96 random clones (without selection) revealed a broad spectrum of transition and transversion mutations with no significant nucleotide bias.

We then electroporated the resulting 4 independent ligation reactions into competent cells of strain SQ765 (genotype: MG1655, ΔhsdR) (*Table 1*) to generate four independent mutant plasmid libraries, each containing ~$10^6$ clones. We then extracted the plasmid from these four independent libraries that had been constructed in SQ765 and transformed the plasmids into the *spy* and *ampC* null strain SQ2041, which contains pBR322bla::GSlinkerIm7 L53A I54A. The transformants were plated onto plates containing 4 mg/ml penicillinV, supplemented by 0.1 mM IPTG to induce Spy, and incubated at 37°C overnight. We picked 159 penicillin V resistant colonies and streaked them twice to

obtain single colonies. These cells were further tested for penicillin V resistance by spot titration onto plates with increasing amounts of penicillin V (3, 4, and 5 mg/ml) supplemented with 0.1 mM IPTG, using strain SQ2068 as a negative control. 65 of the 159 strains consistently showed resistance to the three penicillin V concentrations; plasmids were extracted from these 65 strains and sequenced to identify mutations in the mature protein-encoding region of Spy.

## Measurement of the in vitro activity of the Spy variants

Activities of the Spy variants were assayed for their ability to prevent the aggregation of two clients: chemically denatured aldolase and reduced denatured α-lactalbumin (α-LA), as well as their ability to assist the refolding of chemically denatured aldolase.

The aggregation of bovine α-lactalbumin (type III from Sigma Aldrich, St. Louis, MO) was initiated by reducing its disulfide bonds with DTT at 25°C as previously described (*Kulig and Ecroyd, 2012*) in the presence or absence of Spy. Each reaction contains 50 µM of α-lactalbumin, 0-100 µM of Spy, and 20 mM DTT in a buffer composed of 50 mM sodium phosphate, 100 mM sodium chloride, and 5 mM EDTA, pH 7.0. 100 µl of each reaction solution was incubated in an acrylic UV transparent flat-bottom 96-microwell plate (Corning, Corning, NY) sealed with a transparent film (Denville Scientific, Inc. South Plainfield, NJ). The light scattering light due to the aggregation of reduced α-lactalbumin was monitored at 360 nm using a Synergy HT Multi-Mode Microplate Reader (Biotek, Winooski, VT) with readings taken every 15 min for 10 hr at 25°C following an initial period of medium speed shaking for 30 s before each reading. Spy variants were added to the aggregation reaction at three concentrations: 5, 7.5, and 10 µM to make the final Spy: α-lactalbumin ratios equal to 0.1, 0.15, and 0.2, respectively. Light scattering data at four endpoints (300, 400, 500, and 600 S) were used to calculate the relative activities of different Spy variants based on standard curves generated with 22 different Spy: α-lactalbumin ratios. The average of the 12 relative activity values (three concentrations each calculated according to four standard curves) of each mutant was then calculated and standard errors (from the calculation for average) are given. Typical aggregation curves and standard curves can be found in *Figure 5—figure supplement 2*. Quantification was carried out as follows: to measure the relative activity of Spy variants on α–LA, each Spy variants were used at three ratios in duplicate (Spy: α–LA ratios = 0.1, 0.15, and 0.2) in the aggregation assay. The standard curve was smooth and could have been usable over a broad range. These specific ratios were chosen because they A) fell in a steep portion of the standard curve so that the light scattering signal is the most sensitive to the amount of Spy added and because B) the Spy concentrations at these ratios were high enough to enable pipetting high enough volumes to minimize pipetting errors. The portion of the standard curve we used is indicated by the red box in *Figure 5—figure supplement 2*. Light scattering endpoints at 300, 400, 500, and 600 S were taking from each aggregation curve and then back-calculated according to the standard curve at the respective time point to get the effective ratio of Spy: α–LA (or, ratio equivalent to Spy WT: α–LA). The activity of Spy variants relative to wild type is then calculated from dividing these effective ratios by the adding ratios. For example, Q100L added at a Spy: α–LA ratio of 0.2 had a light scattering reading of 0.009 arbitrary units (a.u.) after 400 S of the reaction. According to the standard curve at 400 S, this is equivalent to wild type Spy added at a Spy:α–LA ratio of 0.383. Therefore, Q100L has the activity 1.82-fold of wild type based on the calculation from the 400 S standard curve. This calculation is then repeated using the other three standard curves and other light scattering endpoint values taken from the aggregation curves of the same variant added at different Spy:α–LA ratios. This strategy allowed us to generate 12 relative activity values (in duplicate) of the same variant, which are then averaged to give the final value. The standard errors we reported reflect the average of these 24 activity measurements. To validate our calculations, we performed the aggregation assay with wild type Spy in exactly the same way as we used for the Spy variants. This gave us a value of 1.06 ± 0.09 fold activity relative to wild type Spy, suggesting that our approach to compare the activities of the Spy variants to wild type is precise and reproducible.

Ammonium sulfate suspension of aldolase from rabbit muscle (MP Biochemicals, Santa Ana, CA) was suspended in a 40 mM HEPES, 150 mM NaCl pH7.5 buffer that contained 4 mM Beta-mercaptoethanol and then dialyzed into 40 mM HEPES, 150 mM NaCl pH7.5 buffer. 100 µM aldolase was then denatured in 40 mM HEPES pH 7.5 with 50 mM NaCl, 3 M guanidine, and 2 mM DTT for >2 hr at room temperature. Aggregation of denatured aldolase was initiated by rapidly diluting 6.5 µl of the denatured aldolase into 1293.5 µl (a 200-fold dilution) of 150 mM sodium chloride pH 7.5 buffer pre-equilibrated for 10 min at 23°C. The final aldolase concentration reached was 500 nM. The aggregation of aldolase was

monitored by measuring light scattering in the absence or presence of Spy variants with a photomultiplier Spectrofluorimeter (Photon Technology International (PTI), Birmingham, NJ) and FeliX32 Analysis software (PTI, inc.). with excitation/emission wavelengths and slits of 360 nm and 1 nm respectively. We found that pipetting the aldolase into the cuvette at the same position relative to the stirring bar made for a very reproducible assay. Thus, long gel loading tips (83 mm, MultiFlex Pipet Tips from Thermo Fisher Scientific, Waltham, MA) were used to pipette denatured aldolase into the cuvette through the small sample injection hole on the top of the lid of the PTI photomultiplier Spectrofluorimeter, so that for each injection the end of the tip lay just above the stirring bar. We used different ratios of wild Spy: aldolase to establish a standard curve. For most of the Spy variants we used a ratio of Spy: aldolase of 0.3 because this lies in the middle of the standard curve. Our most active variant Q25R, completely suppressed aldolase aggregation at this ratio so it was in addition further diluted and measured at a Spy: aldolase ratio of 0.1. The aggregation rates were measured by calculating the slopes of the aggregation curves (fitting the data points between 50 s and 600 s of the aggregation reaction). Each Spy variant was measured three times. By using the standard curve, we were able to calculate the relative activity of these Spy variants as shown in *Figure 5—figure supplement 3*. The error of the variant's measurements and the error of the standard curve were combined to give a final error.

Aldolase used in the refolding assay was prepared by dissolving powdered aldolase (Cat # 0215985925, MP Biochemicals, USA) into 40 mM HEPES, 150 mM NaCl, pH7.5 buffer. Aldolase was then denatured in 40 mM HEPES pH 7.5 with 50 mM NaCl, 3 M guanidine, and 2 mM DTT at 25 µM overnight at room temperature. To initiate refolding of aldolase, denatured aldolase was diluted 100-fold into the refolding buffer (150 mM NaCl, 40 mM HEPES, 5 mM DTT, pH7.5) to a final concentration of 0.25 µM in the absence or presence of different ratios of Spy at 25°C. After 5 min, 14 µl aliquots from the refolding reaction were added into 200 µl assay buffer to test for aldolase activity at 25°C. The assay buffer contained 0.15 mM β-nicotinamide adenine dinucleotide reduced disodium salt (Sigma Aldrich, USA), 2 mM Fructose 1,6-diphosphate (Sigma Aldrich, USA), 0.18 U/ml α-glycerophosphate dehydrogenase/triosephosphate isomerase (Sigma Aldrich, USA), 150 mM NaCl and 40 mM HEPES, pH 7.5. The measurements of absorbance at 340 nm were immediately started using a Synergy HT Multi-Mode Microplate Reader (Biotek, Winooski, VT) following a 5-s shaking at medium speed. Data were collected for 5 to 10 min and the absorbance values were plotted against time to obtain a slope, which was then divided by the value corresponding to 100% activity (*i.e.*, that of an equivalent concentration of native aldolase) to obtain the percentage of native aldolase. Using Spy (wt): aldolase ratios of 0,0.125,0.25,0.5, 0.75, 1, 1.25, 1.5, and 1.75, we obtained a standard curve shown in *Figure 5—figure supplement 4* and found that the ratio of 0.25 Spy:Aldolase was located in the middle of the range. Thus we performed refolding assay at a Spy: aldolase ratio of 0.25 for all the Spy variants and calculated their relative activities in assisting aldolase refolding according to the standard curve (see *Figure 5—figure supplement 4* legend for details). Every Spy mutant was measured at least three independent times and the error of the measurements and the error of fitting the standard curve were combined to give a final error.

## Quantitative western blotting

The steady-state expression levels of the different Spy variants and the β-lactamase-Im7 L53A I54A biosensor in strains LW53-LW59 (*Table 1*) were quantified by western blotting of whole cell extractions using infrared fluorescence labeled IRDye secondary antibodies. This was done after induction by various amounts of IPTG (0.01–0.5 mM) for 3 hr. IRDye 680LT goat anti-mouse secondary antibody (LI-COR Biosciences, Lincoln, NE) was used to recognize the primary antibodies, which were against either the β-lactamase portion of the biosensor or directed against Spy. As a loading control, we quantified the amount of maltose binding protein (MBP) present in the lysate using a IRDye 800CW goat anti-rabbit secondary antibody (LI-COR Biosciences) directed against MBP. 0.04% maltose was added to the medium to induce MBP expression. MBP is a periplasmic protein that is routinely used as a loading control (*Raivio et al., 1999*); it has the added advantage that it is not a Spy client (unpublished results). Various dilutions of the whole cell extracts ($A_{600}$ = 2.5) were loaded and their intensities were plotted to quantify the level of Spy and β-lactamase-Im7 L53A I54A biosensor protein in the Spy variants.

## Calculation of specific in vivo activity of Spy variants

We performed spot titrations of strains expressing seven different super-Spy variants as well as the β-lactamase-Im7 L53A I54A biosensor onto various concentrations of penicillin V (0 through 7000 µg/ml)

and IPTG (0, 0.01, 0.1, 0.2, and 0.5 mM) to quantitatively measure the minimal inhibitory concentration (MIC) they exhibited toward the β-lactam antibiotic penicillin V using the methods previously described (*Foit et al., 2009*). To measure the specific in vivo activity of these Spy variants, we then normalized the maximal MIC values of each strain to the levels of Spy expressed in these variants to compensate for possible differences in Spy expression. The maximal MIC values were achieved at 0.5 mM IPTG induction for all the variants except for Q49L and WT which reached a maximal MIC starting at 0.2 mM. The steady-state protein levels of each Spy variant expressed at various IPTG concentrations were measured using quantitative western blots as described above. The in vivo specific activity was obtained by calculating the ratio between the maximal MICs obtained for the variant divided by the MIC obtained for wild-type Spy when it was expressed to the same level as the variant. These values are reported as normalized MIC values ($MIC_{norm}$) and are a measure of the in vivo specific activity of the Spy variants.

## Hydrogen-deuterium exchange

Hydrogen-deuterium exchange of Spy and Spy-Im7 complex was performed using top-down ECD-FTICR-MS, as described previously (*Pan et al., 2009*, *2010*; *Serpa et al., 2013*). This approach relies on the rapid scrambling-free fragmentation of the intact protein by electron capture dissociation (ECD). The deuteration level of each amino acid residue then can be deduced from the c- and z-ion series produced. Briefly, Spy and Im7 protein stock solutions (250-1000 μM) in 40 mM HEPES, 150 mM NaCl, pH 7.5 were diluted with 10 mM ammonium acetate to 25 μM final Spy (dimer) and 100 μM Im7. Spy-Im7 complex was preformed in a 1:4 molar ratio of Spy dimer to Im7. Spy or Spy-Im7 samples were continuously mixed from separate syringes with $D_2O$ in a 1:4 ratio (80% $D_2O$ final) via a three-way tee, which was connected to a 100 μm × 21 cm capillary, providing a labeling time of 10 s. The outflow from this capillary was mixed with a quenching solution containing 0.4% formic acid in 80% $D_2O$ from a third syringe via a second three-way tee, and injected into a Bruker 12 T Apex-Qe hybrid Fourier transform mass spectrometer (Bruker Daltonics, Billerica, MA), equipped with an Apollo II electrospray source. In this analytical format, lower deuteration levels would reflect more protection of the amide protons and formation of hydrogen bonding due to residues involvement in secondary structure elements, interaction, or shielding from the solvent. In-cell ECD fragmentation experiments were performed with an m/z 900-1200 precursor selection range using a cathode filament current of 1.2 A and a grid potential of 12 V as previously described (*Serpa et al., 2013*). Approximately 1200 scans were accumulated over the m/z range 250–2600, corresponding to an acquisition time of approximately 30 min for each ECD spectrum. Deuteration levels of the amino acid residues' amide groups were determined from centroid masses of the c- and z-ion series. The analysis of the deuteration status of c- and z-ion fragment series thus allowed us to localize the residues that exhibited major differences in the protection between free Spy and Spy that is occupied by its client protein Im7.

## Crosslinking

Spy-Im7 protein complex was crosslinked with CyanurBiotinDimercaptoPropionylSuccinimide (CBDPS)-H8/D8 (Creative Molecules Inc., Canada). The crosslinked proteins were digested with trypsin and proteinase K and crosslinked peptides were affinity purified with immobilized avidin, then identified by LC-MALDI MS and MS/MS. Zero-length crosslinking brings additional challenges to the mass spectrometric analysis of the crosslinks as the crosslinked peptides formed do not acquire a specific isotopic signature. To facilitate detection and identification of the Spy-Im7 zero-length crosslinks, we applied a variation of the method using [15]N metabolically-labeled oligomeric proteins (*Taverner et al., 2002*). We used an equimolar mixture of non-labeled and [15]N metabolically-labeled Im7 for crosslinking the Spy-Im7 complex. Following digestion of the crosslinked complex, every Im7 peptide and, most notably, inter-protein Spy-Im7 crosslinks were represented by a pair of light and heavy isotopic forms, derived from non-labeled and [15]N-labeled Im7 peptides, respectively. Mass difference between such pairs is then determined by the number of the nitrogen atoms in the Im7 peptides. We have implemented this principle, as an additional selection criterion, into the algorithm for the automatic detection and identification of the inter-protein crosslinks in heteromeric protein complexes. Technical details are described below:

### CBDPS

16 μl Spy and Im7 7-45, both 26 μM, in PBS pH 7.4, 20 mM $Na_2HPO_4$ were crosslinked with 50 μM of an equimolar mixture of light and heavy isotopic forms of CyanurBiotinDimercaptoPropionyl-Succinimide

(CBDPS-H8/D8; Creative Molecules Inc.) for 30 min at 25°C. The reaction mixtures were quenched with 10 mM ammonium bicarbonate at 25°C. The empty Spy samples (Spy in solution without client) were prepared in the same manner except Im7 7-45 was absent from the reaction mixture.

## ABAS

84 µl Spy and Im7 7-45, both 26 µM, in PBS pH 7.4, 20 mM $Na_2HPO_4$ were crosslinked with 100 nM of an equimolar mixture of light and heavy isotopic forms of Azido Benzoic Acid Succinimide (ABAS-$^{12}$C6/$^{13}$C6; Creative Molecules Inc.) for 30 min at 25°C followed by 10 min UV irradiation from the top of an open 0.2 ml reaction tube with a 25W 254 nm UV lamp (~2 cm distance). The reaction mixtures were quenched with 10 mM ammonium bicarbonate at 25°C.

## PICUP

Photo-Induced Cross-Linking of Unmodified Proteins (PICUP) crosslinking: 18 µl 26 µM Spy and 26 µM equimolar mixture of non-labeled and $^{15}$N metabolically-labeled Im7 7-45 in PBS pH 7.4, 20 mM $Na_2HPO_4$ were supplemented with 50 nM tris(2,2'-bipyridyl)ruthenium (II) dication and 1 mM ammonium persulfate. Crosslinking was induced by exposing the reaction mixture in a clear Eppendorf tube to 30 rapid LED flashes. The reaction mixtures were quenched with 10 mM 2-mercaptoethanol.

## EDC

18 µl of 26 µM Spy and 26 µM equimolar mixture of non-labeled and $^{15}$N metabolically-labeled Im7 7-45 in PBS pH 7.4, 20 mM $Na_2HPO_4$ were crosslinked using 30 mM 1-ethyl-3-(3-dimethylaminopropyl) carbodiimide (EDC) for 30 min at 25°C. The reaction mixtures were quenched with 10 mM ammonium bicarbonate at 25°C.

Crosslinked proteins were digested with either trypsin (Promega, Madison, WI) for 18 hr at 37°C or proteinase K (Worthington Biochemical Inc., Lakewood Township, NJ) for 60 min at 37°C, both at 1:15 (wt:wt) enzyme:client ratios. Trypsin and proteinase K digestions were inhibited by the addition of 4-(2-Aminoethyl)benzenesulfonyl fluoride hydrochloride (AEBSF) to a final concentration of 10 mM.

CBDPS crosslinked peptides were enriched on monomeric avidin beads (Thermo Fisher Scientific), eluted from the beads with 0.1% TFA 50% acetonitrile, concentrated by lyophilization, reduced with 25 mM DTT for 10 min at 25°C, and finally acidified to pH 2 with formic acid. ABAS, PICUP, and EDC samples were reduced with 25 mM DTT for 10 min at 25°C and subsequently acidified to pH 2 with formic acid.

Mass spectrometric analysis was carried out with a nano-HPLC system (Easy-nLC II, ThermoFisher Scientific) coupled to the ESI-source of an LTQ Orbitrap Velos mass spectrometer (ThermoFisher Scientific), as described earlier (*Tonkin et al., 2013*). Samples were injected onto a 100 µm ID, 360 µm OD trap column packed with Magic C18AQ (Bruker-Michrom, Auburn, CA), 100 Å, 5 µm pore size (prepared in-house) and desalted by washing for 15 min with 0.1% formic acid (FA). Peptides were separated with a 60 min gradient (0–60 min: 4–40% B, 60–62 min: 40–80% B, 62–70 min: 80% B, with solvent B: 90% acetonitrile, 10% water, 0.1% FA) on a 75 µm ID, 360 µm OD analytical column packed (in-house) with Magic C18AQ, 100 Å, 5 µm pore size with IntegraFrit (New Objective Inc., Woburn, MA) and equilibrated with 95% solvent A (2% acetonitrile, 98% water, 0.1% FA).

MS data were acquired with Xcalibur (version 2.1.0.1140) with Mass Tags and Dynamic Exclusion precursor selection methods enabled in global data dependent settings. For CBDPS-H8/D8 and ABAS-$^{12}$C6/$^{13}$C6, mass differences between light and heavy isotopic forms of 8.05 and 6.02 Da were used in the Mass Tags setting, respectively. For PICUP and EDC experiments using $^{15}$N metabolically-labeled Im7, mass differences in the Mass Tags acquisition method were set according to all the possible nitrogen atoms in Im7 tryptic peptides. MS scans and MS/MS scans were acquired in the Orbitrap mass analyzer at 60,000 and 30,000 resolution, respectively. MS/MS fragmentation was performed by CID activation at a normalized collision energy of 35%. Data analysis was performed using DXMSMS Match of ICC-CLASS (*Petrotchenko and Borchers, 2010*) and 14N15N DXMSMS Match.

## Bio-layer interferometry

Im7 L53A I54A was biotinylated by incubation with EZ-link NHS-biotin (Thermal Fisher Scientific) at 1:1 molar ratio (100 µM each) in the assay buffer (40 mM HEPES, 150 mM NaCl, pH7.5) for 30 min at room temperature. Then the reaction was quenched by addition of one tenth volume of 1M Tris pH7.5 and the unconjugated biotin was removed by dialyzing in the assay buffer overnight at 4°C. Binding kinetics were performed on the Octet RED system (Fortebio, Menlo Park, CA) at 25°C. The streptavidin sensors

were pre-wetted in the assay buffer for 15 min before use. Spy variants were serial diluted to 2, 1, 0.5, 0.25, and 0.125 µM in a 96-well plate in the assay buffer. The binding assay contains the following steps: immobilization of the biotin-conjugated Im7 L53A I54A (10 µg/ml) for 20 min, wash for 5 min, baseline for 1 min, association for 10 min, and dissociation for 15 min. The last three steps were repeated for all the Spy concentrations. Wells with assay buffer only were used as reference wells and were subtracted from the raw data. Curve fitting was performed in Sigmaplot assuming a 1:1 binding model using the following equations for association and dissociation, respectively: $y = a * \left(1 - e^{-(k_{obs} * t)}\right) + y_0 * t$ and $y = y_0 + a * e^{-(k_{off} * t)}$, where $y_0$ is the constant to correct for baseline drift. The observed association rate constants ($k_{obs}$) are then plotted against Spy concentrations to obtain the association rate constants ($k_{on}$) according to the following equation: $k_{obs} = k_{on} * [Spy] + k_{off}$. The dissociation constant KD is calculated according to: $KD = \dfrac{k_{off}}{k_{on}}$.

## Limited proteolysis

Digestion of Spy alone (50 µM) or Spy-Im7 L53A I54A complex (50 µM each) was carried out at 1:100 mass ratio of trypsin to protein in 40 mM HEPES, 100 mM NaCl, pH 7.5 at room temperature. At different time points (0–8 min), aliquots were withdrawn and the digestion was stopped with 10% TFA. Peptides were separated by a reverse phase C18 column (Zorbax 300SB-C18, 1 × 50 mm, 3.5 µm, Agilent, Santa Clara, CA) at room temperature and then applied to a Q-TOF dual ESI LC/MS (Agilent) for identification. A 15 min linear gradient of 2–80% acetonitrile in 0.1% formic acid at a flow rate of 0.3 µl/min was used to elute the peptides. Peptide identification was performed using BioConfirm software (Agilent).

## Identification of Im7 L53A I54A 7-45 peptide

To identify the minimal length peptide derived from Im7 L53A I54A that mediates the binding to Spy, we applied limited proteolysis to Im7 L53A I54A with trypsin to prepare a series of incompletely digested fragments. After quenching the digestion reaction, the peptide mixture was mixed with the N-terminal strep-tagged Spy and applied to a strep-tactin sepharose column (IBA, Germany). The complex was eluted by 2.5 mM desthiobiotin, separated by a reverse phase C18 column (Zorbax 300SB-C18, 1 × 50 mm, 3.5 µ) at room temperature, and then applied to a Q-TOF dual ESI LC/MS (Agilent) for identification. We obtained fragments 5–43, 5–70, 5–73 and 5–76 from Im7 L53A I54A. The shortest fragment (5–43) constitutes most of the α1 and α2 helices in Im7. We then screened a few constructs containing extension or deletion at the N or C termini with amino acid sequences derived from the Im7 protein to identify the most stably expressed fragment in vivo. We purified three peptides with the highest expression levels (3–45, 7–45, and 12–45) and characterized their interactions with Spy by isothermal titration calorimetry. Based on their affinities to Spy ($K_d$ = 2.5, 2.3, 2.6, and 8 µM for the full length Im7 L53A I54A, 3–45, 7–45, and 12–45 peptide, respectively) and their expression levels, we chose the 7–45 fragment as the minimal length peptide that represents the interaction between Spy and Im7 L53A I54A.

## NMR spectroscopy

A $^{15}$N-HSQC experiment was performed on an Agilent 600 MHz spectrometer equipped with a triple-resonance cryoprobe. The sample contained 200 mM $^{15}$N labeled Im7 7-45 dissolved in 13 mM MES, 13 mM HEPES, 50 mM NaCl pH7.0, supplemented with 10% $D_2O$.

## Spy/Im7 L53A I54A docking

The B-factors of residues Phe29-Asn33 in the crystal structure of Spy (PDB ID: 3O39) are very high (>110), meaning that these residues are possibly flexible. We therefore deleted these residues from the structure before docking analysis because they limit the movable space of the Im7 peptide around the Spy structure. The 3D structure model of Im7 was obtained using the iterative threading assembly algorithm I-TASSER (*Zhang, 2008*; *Roy et al., 2010*). We also used the *ab inito* structure modeling algorithm QUARK (*Xu and Zhang, 2012*) to model the structure of Im7. Both the QUARK and I-TASSER models of Im7 contain two α-helices, but because I-TASSER modeling has a higher overall confidence score, it was used for docking.

A three-step hierarchical approach was used to dock Im7 against Spy:

### Generation of initial docking poses

ZDOCK (*Chen et al., 2003*) was used to perform *ab initio* docking search in which the Spy structure was fixed and the Im7 structure was translated and rotated around the Spy structure. By

default, 1000 docking decoys were generated, which were then clustered based on the distance between the geometric centers of the Im7 decoy structures using a distance cutoff of 4 Å. The top 20 docking models ranked by the docking score were selected from the biggest cluster for further analysis.

### Optimization and refinement of docking poses

Starting from the top ZDOCK models, ModRefiner (*Xu and Zhang, 2011*) was used to refine the atomic details of structural models and to improve the steric clashes that are often found from the rigid-body docking models.

### Final model selection

The final complex model was selected from the refined structures based on the energy score in ModRefiner (*Xu and Zhang, 2011*). The selected model has energy below −11,000. We then manually added the Lys30-Asn33 to the N-terminal ends of the Spy dimer.

## Sequence alignment

10 rounds of PSI-BLAST (*Altschul et al., 1997*) using the *E.coli* K12 *spy* sequence were performed to obtain 503 Spy homologous protein sequences. To separate the sequences of Spy orthologs from the sequences of their homolog, CpxP, the 503 sequences were aligned in SeaView (*Gouy et al., 2010*) using the built-in algorithms MUSCLE (*Edgar, 2004*) and MAFFT (*Katoh et al., 2002*), and then these results were used to construct a phylogenetic tree using MEGA5 (*Tamura et al., 2011*). 252 sequences were assigned to the Spy clade and the rest were assigned to the CpxP clade. The sequences were then aligned with ClustalW (*Thompson et al., 2002*).

## Stability measurement for Spy variants

The thermodynamic parameters characterizing the unfolding of the Spy variants were derived from analysis of the thermal denaturation curves, which were obtained using circular dichroism measurements at 222 nm to follow unfolding. Spy variants were diluted to 6.5 µM in 10 mM sodium phosphate buffer (degassed), pH7.5 and were denatured at a heating rate of 2°C/min from 10°C to 80°C in a 1 mm path length quartz cuvette. The circular dichroism signal was monitored by a JASCO J-810 CD spectrometer (JASCO inc., Easton, MD) equipped with a Peltier thermoelectric controller. Samples were quickly cooled down to 10°C at the end of the denaturation reaction and each spectrum after renaturation was compared with the original spectrum at 10°C to check the reversibility of the denaturation reactions. All the Spy variants after renaturation gained at least 95% of the initial CD signal, indicating that the denaturation process is reversible for all the variants under the conditions we have used. Data analysis was performed in Sigmaplot according to equation II as described in the supplemental materials in (*Greenfield, 2006*) to obtain the melting temperature ($T_m$) and the enthalpy change ($\Delta H_m$) of each denaturation reaction at pH7.5. Each thermal denaturation measurement was performed three times to get the average values of $\Delta H_m$ and $T_m$ and the standard errors (*Table 2*). To find the heat capacity change ($\Delta Cp$) when the protein unfolds we obtained Tm and $\Delta H_m$ as a function of pH by measuring the denaturation curves at a wide range of pH from 2.0 to 10.0 in 20 mM sodium phosphate buffer. We then plotted the $\Delta H_m$ against Tm to yield $\Delta Cp$ (*Table 2*), which is the slope. Finally, the unfolding free energy change $\Delta G_{NU}$ at 25°C is calculated using the modified Gibbs-Helmholtz equation according to *Grimsley et al., (2013)*:

$$\Delta G(T) = \Delta H_m \left(1 - T / T_m\right) - \Delta C_P \left[\left(T_m - T\right) + T ln(T / T_m)\right]$$

where T = 298.15K.

## Acknowledgements

We thank Philipp Koldewey, Linda Foit and Ursula Jakob for useful discussions, Jingxi Pan for assistance with acquisition of the HDX data and Logan Ahlstrom with assistance in doing the electrostatic calculations shown in *Figure 4—figure supplement 2A* and in preparation of this figure. The authors at the UVic-Genome BC Proteomics Centre (EVP, KATM and CHB) would like to thank Genome Canada, Genome BC, and the Western Economic Diversification of Canada for platform funding and support. JCAB is a Howard Hughes Medical Institute investigator.

## Additional information

### Funding

| Funder | Grant reference number | Author |
| --- | --- | --- |
| Howard Hughes Medical Institute | | Shu Quan, Lili Wang, Scott Horowitz, James CA Bardwell |
| Genome Canada | | Evgeniy V Petrotchenko, Karl AT Makepeace, Christoph H Borchers |
| Western Economic Diversification of Canada | | Evgeniy V Petrotchenko, Karl AT Makepeace, Christoph H Borchers |
| National Institute of General Medical Sciences | GM083107, GM084222 | Yang Zhang |

The funders had no role in study design, data collection and interpretation, or the decision to submit the work for publication.

### Author contributions

SQ, EVP, Conception and design, Acquisition of data, Analysis and interpretation of data, Drafting or revising the article; LW, KATM, JY, Acquisition of data, Analysis and interpretation of data; SH, Acquisition of data, Analysis and interpretation of data, Drafting or revising the article; YZ, CHB, Conception and design, Analysis and interpretation of data; JCAB, Conception and design, Analysis and interpretation of data, Drafting or revising the article

## Additional files

### Major dataset

The following previously published dataset was used:

| Author(s) | Year | Dataset title | Dataset ID and/or URL | Database, license, and accessibility information |
| --- | --- | --- | --- | --- |
| Quan S, Koldewey P, Tapley T, Kirsch N, Ruane KM, Pfizenmaier J, Shi R, Hofmann S, Foit L, Ren G, Jakob U, Xu Z, Cygler M, Bardwell JC | 2011 | Periplasmic protein related to spheroblast formation | 3O39; http://www.rcsb.org/pdb/explore/explore.do?structureId=3O39 | Publicly available at the RCSB Protein Data Bank (http://www.rcsb.org/). |

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
