## [Decision Letter]

Thank you for sending your work entitled “Hyperactive Spy variants implicate flexibility in chaperone action” for consideration at *eLife*. Your article has been favorably evaluated by a Senior editor and 3 reviewers, one of whom is a member of our Board of Reviewing Editors.

The Reviewing editor and the other reviewers discussed their comments before we reached this decision, and the Reviewing editor has assembled the following comments to help you prepare a revised submission.

The three reviewers were in agreement that this is an interesting study exploring stability, conformational disorder, and chaperone activity. The authors have enlisted a variety of techniques to address these issues and overall the paper is well written and thought provoking. The reviewers did have some concerns, however, that need to be addressed.

1) Please explain why you only screened for and studied mutants with increased activity. In fact, in the Discussion you state you're surprised that all Spy variants were more active in aggregate prevention of Aldolase and α-Lactalbumin, but why is this a surprise since they only selected for improved chaperone activity. Are there other examples where mutant chaperones had improved activity on one client and reduced activity on other clients?

2) Somewhat related to point 1: looking at Table 2, the best mutant Spy chaperones for lm7 do not seem to be the best for aldolase or α-lactalbumin. E.g., the Q100L mutants shows the largest effect for lm7, whereas it only marginally improves Spy's chaperone activity for Aldolase. For Aldolase, the Q25R is the best chaperone, but this mutant actually shows the weakest increase for lm7. What does this imply?

It would thus be interesting to provide similar analyses as done in Figure 6 for lm7 for these 2 clients as well to see what can be deduced from such analysis and to see how valid the general statements made in the discussion (primarily based on lm7) hold true.

3) Data on other “canonical” chaperones have always suggested that one requires a dynamic cycle to modulate client affinity regulated by ATP driven reactions. Intriguingly, this seems not the case for Spy. Here, the authors even show that increasing affinity is even linearly associated with further increase client affinity. Would one not expect a bell-shaped curve here? I.e., for mutants with the highest affinity, client off rates should be so low that, whilst aggregation prevention may still be high, client release would be so low that efficient client folding would no longer occur. In fact, it would be highly informative to also add data on protein refolding activity of these mutants. It would be appreciated if the authors could elaborate on this to some extent in their Discussion.

4) There are some confusing aspects of the study and figures that are at odds. For example, it's confusing that much is made of the hydrophobic patches on Spy (P1 and P2 in Figure 1). Three of the variants increase and connect the patches (Figure 1), creating, presumably, a more favorable binding surface for unfolded or partially unfolded proteins. This is quite convincing. Then the probability calculations in Figure 1 are at odds with this by predicting that the highest probability sites for the interface are not in the hydrophobic binding patches. A program called ProMate was used. No details were provided, including the reference, so it's difficult to evaluate the method and reliability of the predictions but they are logically at odds with the other rationalizations for activity and their other figures, as mentioned. Along similar lines Figure 4—figure supplement 2, panel A is confusing with respect to Figure 1 and the ‘story’ presented in the manuscript. There are two hydrophobic patches on Spy, which turn into one large hydrophobic surface upon mutation, yet panel A shows essentially that entire surface to be positively charged. I don't see how both can be true. You mention using a plugin in Pymol (no reference, no details). My guess is that the electrostatic potential wasn't calculated correctly or without knowledge of thresholds and solvent screening methods. Nonetheless the logical inconsistencies between different figures and the text make for a confusing read. In general the various computational tools seem to have been naively applied to the problem. These things need to be addressed, and if there are real conflicts then they need to be discussed but they aren't addressed here, or maybe the inconsistencies weren't noticed, or perhaps the figure legends are inaccurate.

5) Then there are two other things of concern. One is that the main claim is that Spy becomes more active when it is more flexible/unfolded yet the modeling is done with static WT Spy crystal structures. The second is related, the experiments that are equated with flexibility measurements (hydrogen exchange and protease susceptibility) were performed on only WT Spy according to Figure 2. I don't see results for 'flexibility' experiments for the variants, particularly Q100L, H96L and Q49L. Given the authors' argument that flexibility is linked to activity (and increased activity is demonstrated) I think some results relating to the increased flexibility of the variants (limited possibly just to the main 3 mutants) should be explicitly provided. Then there is another issue that raises questions, it isn't clear that the protease experiments are reliable or even support the binding and 'flexibility' arguments. For example, in looking at Figure 2—figure supplement 2 the most striking thing is that most of the faster cleavage times are for the Spy Im7 complex (tall magenta bars). This is very confusing and at odds with the proposed binding models. The authors do point out the inconsistency but then essentially ask us to ignore it and focus on only the cases where the magenta bars are low (i.e., higher incubation times or > 8 min). I assume that if there were a structural explanation from the models that it would have been presented.

6) The title of the paper is ‘Hyperactive Spy variants implicate flexibility in chaperone action’. The results are all indirect with respect to flexibility. For example, increased hydrogen exchange is taken as proof of flexibility, but the method does not measure flexibility per se. For example, you can have a rigid, stiff extended chain that exchanges 100% and to say the chain is flexible would be wrong. What makes me mention this is that most of the results are presented in a very direct way with respect to the wording by providing inference as fact or by using the word indicate or determine instead of using the word suggest. While this may seem like a fine point, much of the manuscript reads as something of a house of cards that is strongly presented as fact. It is recommended that the manuscript is edited to tone some of this down.

---

## [Author Response]

*1) Please explain why you only screened for and studied mutants with increased activity*.

We exclusively selected for activity enhancing mutations because we consider them to be much more informative as to chaperone mechanism than loss of function mutants. This is because there are in general many more ways of disrupting function than enhancing it, and many of the ways of disrupting function are not particularly informative. Rather than individually mutating residues and testing for loss of function, one can simply exploit what evolution has done to a protein to gain a good idea of which residues are important for the function by simply looking at residue-by-residue conservation. We have clarified our rational for looking at activity enhancing mutations by adding these sentences to the results section: “Our ability to link protein folding to antibiotic resistance gives us a unique opportunity to select for activity enhancing mutations in a chaperone. Analysis of the reasons behind the improved chaperone ability of activity enhancing mutants of Spy should inform us about Spy’s catalytic mechanism and perhaps also tell us what makes for a good chaperone. We reasoned that activity-enhancing mutations would be more informative in general than those that decreased function, in part because there are a wider variety of uninteresting reasons that mutations can disrupt function such as those causing chain termination.”

*In fact, in the Discussion you state you're surprised that all Spy variants were more active in aggregate prevention of Aldolase and α-Lactalbumin, but why is this a surprise since they only selected for improved chaperone activity*.

We were surprised, but pleased, because we had selected them on the basis of improving the folding of one specific unstable protein but they turned out to be generally improved. Using the geneticists rule of thumb “you get what you select for” we had expected our variants to show an improved ability to fold the substrate they had been selected on but thought it unlikely that they would be generally improved in chaperone activity because we had assumed that evolution had already selected for maximal chaperone ability. Instead, we were thrilled that they showed overall improvement in the ability to inhibit aggregation of the two eukaryotic derived chaperone substrates that we tested.

To discuss the types of mutations that we thought likely to be isolated we added this sentence to the results:

“If we succeeded at all in getting activity enhancing mutations we anticipated obtaining two types of mutations, those that acted in a substrate specific manner that improved the action of Spy only against the substrate for which they were selected on. We also might obtain variants that generally improved the activity of Spy against multiple substrates. If we succeeded in obtaining this latter type of mutations, they should be particularly informative as to what makes a protein an effective chaperone.”

We have also added this explanation to the Results section:

Paragraph beginning: “In genetic selections one usually gets what you select for, thus we had anticipated that the variants we obtained would show an improved ability to refold Im7…”

We are thrilled because a general improvement in chaperone ability makes us more confident that the conclusions we reach are applicable to the generalized chaperone mechanism of Spy rather than some more specialized substrate specific changes (such as say the strengthening of one specific ionic interaction between Im7 and Spy) which would not necessarily tell us much about the generalized chaperone mechanism.

*Are there other examples where mutant chaperones had improved activity on one client and reduced activity on other clients*?

Yes, we had mentioned one example briefly in the submitted manuscript. We have now expanded this as requested and discussed this and another example concerning DnaK in the results section as follows:

Section starting: “…other efforts at improving chaperone activity, though showing some success in generating mutants that were better with the substrates they were selected on, in general showed decreased chaperone activity against other substrates… “

*2) Somewhat related to point 1: looking at*
Table 2*, the best mutant Spy chaperones for lm7 do not seem to be the best for aldolase or α-lactalbumin. E.g., the Q100L mutants shows the largest effect for lm7, whereas it only marginally improves Spy's chaperone activity for Aldolase. For Aldolase, the Q25R is the best chaperone, but this mutant actually shows the weakest increase for lm7. What does this imply*?

It just implies that chaperones show some substrate specificity. As discussed above we expected that a selection to improve the activity for one substrate would result in a decrease in the ability to fold other proteins as had clearly been seen before for GroEL and is the very common result of directed evolution where optimizing for one substrate decreases activity for other substrates.

It would thus be interesting to provide similar analyses as done in Figure 6 for lm7 for these 2 clients as well to see what can be deduced from such analysis and to see how valid the general statements made in the discussion (primarily based on lm7) hold true.

In theory, this would be interesting but it is unfortunately not possible. We are in the fortunate position with Im7 to have a chaperone substrate that is well behaved, most are not and aggregate rapidly making it very difficult to accurately determine on an off rates. To address this comment we have added the following statement to the Results section:

Section starting: “In order to get accurate k_on_ and k_off_ rates we found it is vital to use a substrate that is soluble and not bound to the tip as an aggregate...”

*3) Data on other “canonical” chaperones have always suggested that one requires a dynamic cycle to modulate client affinity regulated by ATP driven reactions. Intriguingly, this seems not the case for Spy. Here, the authors even show that increasing affinity is even linearly associated with further increase client affinity. Would one not expect a bell-shaped curve here? I.e., for mutants with the highest affinity, client off rates should be so low that, whilst aggregation prevention may still be high, client release would be so low that efficient client folding would no longer occur*.

This is a very good point, but we selected for increased in vivo activity so these mutants that lack efficient client release would likely not be obtained in our selection. We have added the following to the Discussion:

Section starting: “Our super-Spy variants demonstrate tighter client binding, and we observe a correlation between the in vitro dissociation constants of these variants to the client protein Im7 L53A I54A and their ability to stabilize Im7 L53A I54A in vivo (Figure 6)...”

*In fact, it would be highly informative to also add data on protein refolding activity of these mutants. It would be appreciated if the authors could elaborate on this to some extent in their Discussion*.

To address the reviewers’ comment we went ahead and tested the abilities of the proteins to refold a substrate. All but one of our variants showed significantly improved abilities or refold aldolase. We have added this date to Table 2 and these sentences to the Results section:

Section starting: “We also tested for the activity of these chaperone variants in their ability to facilitate aldolase refolding. 6 of the 7 of the variants were found to be more active than is wild type Spy in the range of 1.9 to 4.9 fold…”

Because our variants are superior chaperones in multiple assays we are even more confident they are “super” so have changed the title to include this intensifier.

*4) There are some confusing aspects of the study and figures that are at odds. For example, it's confusing that much is made of the hydrophobic patches on Spy (P1 and P2 in*
Figure 1*). Three of the variants increase and connect the patches (*Figure 1*), creating, presumably, a more favorable binding surface for unfolded or partially unfolded proteins. This is quite convincing. Then the probability calculations in*
Figure 1
*are at odds with this by predicting that the highest probability sites for the interface are not in the hydrophobic binding patches. A program called ProMate was used. No details were provided, including the reference, so it's difficult to evaluate the method and reliability of the predictions but they are logically at odds with the other rationalizations for activity and their other figures, as mentioned*.

The Promate predictions actually do very nicely agree with our experimental data and was referenced in the original submission as follows: “Analysis of the Spy crystal structure with the software ProMate (Neuvirth et al., 2004) predicts a potential client binding interface that is overlapping with both hydrophobic patches and the region hit by our mutations (Figure 1)…”

Promate, does not use simple hydrophobicity to calculate binding surfaces. Instead it used a database of the 3D structure of 57 proteins involved in transient protein-protein interactions to generate an interface prediction program. Frankly, there are a host of such prediction programs available, and they give various predictions. To be honest, we prefer experimental data that addresses binding such as those provided by our superspy mutants, and our crosslinking, mass spec and deuterium and protease protection experiments. It is very nice that Promate gave results similar to our experimental data but does not really add much to the paper so we just deleted this sentence and Figure 1.

*Along similar lines*
Figure 4—figure supplement 2*, panel A is confusing with respect to*
Figure 1
*and the ‘story’ presented in the manuscript. There are two hydrophobic patches on Spy, which turn into one large hydrophobic surface upon mutation, yet panel A shows essentially that entire surface to be positively charged. I don't see how both can be true. You mention using a plugin in Pymol (no reference, no details). My guess is that the electrostatic potential wasn't calculated correctly or without knowledge of thresholds and solvent screening methods. Nonetheless the logical inconsistencies between different figures and the text make for a confusing read. In general the various computational tools seem to have been naively applied to the problem. These things need to be addressed, and if there are real conflicts then they need to be discussed but they aren't addressed here, or maybe the inconsistencies weren't noticed, or perhaps the figure legends are inaccurate*.

We apologize for not citing all the bioinformatics programs used. We have now done so. We also thank the reviewers for pointing out the inconsistent standards of displaying hydrophobicity between the figures Figure 1 and Figure 4—figure supplement 2. We consulted about this with Logan Ahlstrom who is an expert in electrostatics calculations. He pointed out that we made two mistakes in our calculations: He says in an email “I think your previous representation could have suffered because 1) you did not include the methionines (the PDB2PQR server for some stupid reason does not recognize the selenomethionines present in the PDB as the methionines they actually are) and 2) the thresholds you used were likely pretty low in magnitude (e.g., +/- 5 KT/e). The tutorial from the APBS site uses +/10 KT/e: http://www.poissonboltzmann.org/apbs/examples/visualization/apbs-electrostatics-in-vmd. This value is largely for visualization purposes, and, at the end of the day, the threshold is really your choice for what highlights the desired properties of your system. From my experience, there is no specific value to use. So I followed what the tutorial did and used +/-10 KT/e.”

Logan kindly redid Figure 4—figure supplement 2 for us and we have used his figure in the resubmitted version. One can now see the hydrophobic patches in both Figure 1 and Figure 4—figure supplement 2 although they are significantly clearer in Figure 1 . This is because the calculation for electrostatic surface used in Figure 4—figure supplement 2 describes the field in the region, considering the impact from the neighboring residues. The hydrophobic surface shown in Figure 1 is colored according to the classification of the individual residues as being hydrophobic or polar; therefore the hydrophobic patches show more clear boundaries in Figure 1.

*5) Then there are two other things of concern. One is that the main claim is that Spy becomes more active when it is more flexible/unfolded yet the modeling is done with static WT Spy crystal structures*.

We have done our modeling starting with the crystal structures of WT Spy because this is what is available. We have shown in Figure 5—figure supplement 1 that although the Spy variants have very different levels of disorder on their own, they achieved similar level of deuterium protection upon peptide binding indicating that they achieved a similar level of order/disorder when they bound to the Im7 7-45 peptide. We therefore think the model based on the crystal structure of WT Spy is also a valid representation for the complex between the Spy variants and the Im7 7-45 peptide.

We have now in at least 3 places made it clear that the Spy-client co-structure model is a tentative model as illustrated by these sentences:

“Given that this model is strongly constrained by only two pairs of residues that were found to interact using the zero-length crosslinkers, it is best regarded as a very tentative and theroretical model. Never-the-less it is consistent with our experimental data and helps in our interpretation of it. The contacts predicted between Im7 and Spy in this tentative model are moderately extensive…”

One could also argue that our models would be more accurate if it was were based on the structure of the Spy mutants; however, obtaining the crystal structure of our Spy mutants would have required a considerable amount of work, and still would not necessarily increase the accuracy of the models. To get an idea of how much the various mutations would affect the overall structure of the Spy-client complex we show below a complex model based on the modeled structure of a theoretical Spy variant that contains all four of the key activity-enhancing mutations (Q49L, H96L, Q100L, F115L) and modeled its binding with the Im7 7-45 peptide. The sequence of this Spy variant was submitted to I-TASSER ([51]; [37]; Roy et al., 2012) to model its structure. The I-TASSER model turns out to be very close to the x-ray structure of wild-type Spy (PDB ID: 3O39) (Figure 7). The TM-score (Zhang and Skolnick, 2004) and RMSD between I-TASSER model of the mutated Spy and the X-ray structure of the wild-type Spy are 0.84 and 1.6 Å, respectively. The Im7 7-45 peptide was docked with the model of the spy mutations and the complex structure was shown in Figure 8. Overall one can see that the structure of this variant is predicted to be not that different from WT Spy. We hasten to add this is a prediction; our experimental data show that the mutants in the absence of client are much less stable than WT Spy making it unlikely that it would be straightforward to solve their crystal structure.Author response image 1.The I-TASSER model of the spy mutations (shown in red cartoon) superimposed onto the X-ray structure of wild-type Spy (shown in grey). Green and blue sticks are the mutated and original residues, respectively.Author response image 2.The modeled complex between Im7 7-45 peptide (blue cartoon) and mutated Spy structure (grey cartoon).

*The second is related, the experiments that are equated with flexibility measurements (hydrogen exchange and protease susceptibility) were performed on only WT Spy according to*
Figure 2*. I don't see results for 'flexibility' experiments for the variants, particularly Q100L, H96L and Q49L. Given the authors' argument that flexibility is linked to activity (and increased activity is demonstrated) I think some results relating to the increased flexibility of the variants (limited possibly just to the main 3 mutants) should be explicitly provided*.

We did do deuterium exchange experiments on all the mutants; this is shown in Figure 5—figure supplement 1 and was described in the original submission in the last paragraph of the Results section as follows:

Section starting: “As another measure of apparent flexibility, we performed deuterium exchange analysis on these Spy variants in the presence and absence of client....”

The deuterium exchange experiment we performed in Figure 2 was analyzed to show different protection to each residue in WT Spy in the presence and absence of the client peptide. We didn't make any conclusions about the overall level of proton protection from the experiments shown in Figure 2.

We did notice, that Q25R, H96L, and Q49L have increased percentage of protected protons indicating they might have more ordered structure compared to wt. Therefore we think gaining structural disorder is one of the reasons that accounts for increased chaperone activity and is not the only reason. Accordingly, we have stated in our manuscript that “Based on our biochemical and biophysical evaluation of the super-Spy variants, there appear to be multiple factors that result in the increased chaperone activity. In addition to increasing binding site hydrophobicity, most of our super-Spy variants decrease Spy’s thermodynamic stability and increase the level of disorder that Spy shows in the absence of client proteins. […] These data suggest that Spy flexibility is important for chaperone function, presumably by facilitating client binding.” And then we go on to discuss why.

*Then there is another issue that raises questions, it isn't clear that the protease experiments are reliable or even support the binding and 'flexibility' arguments. For example, in looking at*
Figure 2—figure supplement 2
*the most striking thing is that most of the faster cleavage times are for the Spy Im7 complex (tall magenta bars). This is very confusing and at odds with the proposed binding models. The authors do point out the inconsistency but then essentially ask us to ignore it and focus on only the cases where the magenta bars are low (i.e., higher incubation times or > 8 min). I assume that if there were a structural explanation from the models that it would have been presented*.

The purpose of this experiment was not to establish the activity-disorder relationship for chaperones (for this purpose one would have to obtain the protease sensitivity of all the Spy variants as a measure of disorder). Instead we used this experiment as an independent way of exploring the potential substrate binding site on Spy. This motivation is clearly stated in the Results section. We agree that limited proteolysis experiment is a low-resolution method to map the potential binding site partly because the preferred cleavage sites are not evenly distributed across the protein sequences. In addition, the amount of protease added to the digestion system played an important role in determining the digestion pattern. We previously found that the addition of a second protein slowed down the digestion of the first protein at almost every site if the amount of protease was kept constant, regardless whether the two proteins interact or not. Therefore, we have decided to keep the ratio of protein: protease constant. This, on the other hand, made the interpretation of “increased cleavage” very difficult, as we explained in the modified figure legend:

“The unstructured termini are more accessible to trypsin in general, and many sites show apparent increased digestion in the presence of client compared to Spy alone...”

*6) The title of the paper is ‘Hyperactive Spy variants implicate flexibility in chaperone action’. The results are all indirect with respect to flexibility. For example, increased hydrogen exchange is taken as proof of flexibility, but the method does not measure flexibility per se. For example, you can have a rigid, stiff extended chain that exchanges 100% and to say the chain is flexible would be wrong. What makes me mention this is that most of the results are presented in a very direct way with respect to the wording by providing inference as fact or by using the word indicate or determine instead of using the word suggest. While this may seem like a fine point, much of the manuscript reads as something of a house of cards that is strongly presented as fact. It is recommended that the manuscript is edited to tone some of this down*.

These are valid points. Most of the measurements we have made imply rather than prove changes in flexibility. For instance, although increased deuterium exchange evidences loss of amide protons protection due to loss of the hydrogen bonding and secondary structure elements, it cannot be directly translated into the increase in flexibility. We have toned down the manuscript throughout, and have substituted the words “apparent flexibility” for “ flexibility”. We have also substituted the word “indicate” with “suggest” in most cases (except for very clear-cut cases. We have also in most cases substituted the weaker word “measured” or “obtained evidence” for “determined”.